# Agents Robust to Distribution Shifts Learn Causal World Models Even Under Mediation

**Matteo Ceriscioli, Karthika Mohan**
School of Electrical Engineering and Computer Science (EECS)
Oregon State University
Corvallis, OR 97331, USA
{ceriscim,karthika.mohan}@oregonstate.edu

## Abstract

In this work, we prove that agents capable of adapting to distribution shifts must have learned the causal model of their environment even in the presence of mediation. This term describes situations where an agent's actions affect its environment, a dynamic common to most real-world settings. For example, a robot in an industrial plant might interact with tools, move through space, and transform products to complete its task. We introduce an algorithm for eliciting causal knowledge from robust agents using optimal policy oracles, with the flexibility to incorporate prior causal knowledge. We further demonstrate its effectiveness in mediated single-agent scenarios and multi-agent environments. We identify conditions under which the presence of a single robust agent is sufficient to recover the full causal model and derive optimal policies for other agents in the same environment. Finally, we show how to apply these results to sequential decision-making tasks modeled as Partially Observable Markov Decision Processes (POMDPs).

## 1 Introduction

Consider an AI system designed to control a sprinkler. The goal of this system is to maintain optimal vegetation conditions while minimizing water consumption. Such a system adapts to seasonal changes and shifts in the rainfall distribution in order to consistently achieve optimal results across diverse weather patterns. This system can be viewed as an agent since it is an entity that maps percepts (e.g. humidity, month of the year) to actions (activation of the sprinkler), and maximizes expected utility (promoting vegetation growth while conserving water) [Russell and Norvig, 1995]. Now, suppose we are tasked with developing an AI system to control a robot responsible for cleaning the windows of a building. One might notice that there is an overlap between the environment in which the sprinkler controller operates and that of the window-cleaning robot. For example, along the Mediterranean coasts of Europe, southerly winds such as the sirocco and libeccio transport dust and sand from the Sahara, which, when mixed with rain, settle and leave persistent stains on glass surfaces. Therefore, meteorological factors such as wind direction and rainfall occurrence also play an important role in the cleaning robot decision task. This raises the question of whether the adaptability of the sprinkler controller entails the acquisition of some form of weather-related knowledge, and whether this knowledge, possibly causal in nature, can be transferred to help us develop robust policies for the window-cleaning robot. In this paper, we show that agents capable of behaving optimally under distribution shifts must indeed possess causal knowledge about their environment, and that this knowledge can be elicited and reused to derive better policies for other agents. Returning to the sprinkler example, its adaptability reflects causal knowledge about the weather, which can be extracted and used to improve the policy of the window-cleaning robot.

Investigating the link between causal understanding and robustness to shifts in the environment is essential for building AI systems capable of generalizing across diverse scenarios [Pearl, 2018,

39th Conference on Neural Information Processing Systems (NeurIPS 2025).

Schölkopf, 2022]. While traditional machine learning excels at pattern recognition within fixed distributions, they often struggle when faced with distribution shifts or interventions that alter the underlying system dynamics. This issue has been extensively studied through diverse methodologies including domain adaptation [Ben-David et al., 2006], transfer learning [Pan and Yang, 2010, Zhuang et al., 2021], federated learning [Konečný et al., 2016], and transportability [Pearl and Bareinboim, 2011], each addressing distinct flavors of the problem. Modern AI systems are expected to meet several key requirements, including robustness to distribution shifts, reliable generalization, transparent decision-making, and avoidance of unintended consequences [Amodei et al., 2016, Hendrycks et al., 2021, Shah et al., 2025]. Causal models offer a powerful framework that addresses these challenges by providing a formal representation of the mechanisms governing an environment [Pearl, 2009]. Causal modeling enables AI systems to generalize more effectively by capturing the underlying causal relationships that persist across scenarios [Schölkopf et al., 2021], and it enhances system explainability by supporting both interventional and counterfactual reasoning [Bareinboim et al., 2024]. Yet, a fundamental question remains: Is causal knowledge truly necessary for generalization, or can agents achieve robustness without causal understanding of their environment?

Recent work [Richens and Everitt, 2024] has demonstrated that agents capable of adapting to distribution shifts must have learned a causal model of their environment, i.e., if the environment can be represented by a causal model, the model's causal structure can be reconstructed by querying an agent capable of behaving optimally under distribution shifts. However, these results focus on single-agent, non-sequential tasks, and rely on the strong assumption of no mediation [Pearl, 2009], meaning the agent's actions cannot have an effect on the utility via environment states. In contrast, many real-world AI applications involve tasks where mediation exists. For example, an autonomous car navigating from point A to point B, may affect lane occupancy and, in turn, traffic flow and the behavior of other drivers.

The goal of this paper is to extend this theoretical framework to the more general mediated case and explore the implications for multi-agent systems and sequential decision tasks. See Appendix A.4 for a comparison with Richens and Everitt [2024]. This work makes the following key contributions:

1. We demonstrate that agents robust to distribution shifts must have learned a causal model of their environment, even when the no-mediation assumption in Richens and Everitt [2024] is relaxed.

2. We present an algorithm to learn the Causal Influence Diagram (CID) depicting mediated decision tasks, by querying optimal policy oracles. We outline how to incorporate prior knowledge into the causal model (Section 3.1).

3. We offer insights into the implications of our findings for multi-agent environments (Section 3.2) and for sequential decision tasks modeled with Partially Observable Markov Decision Processes (POMDPs) (Section 3.3).

## 2 Preliminaries

We are interested in representations of decision tasks that capture both the independence structure and the causal relationships among environmental variables, while explicitly specifying the agent's role, including its observable inputs and the variables subject to its control.

Causal Influence Diagrams (CIDs) [Heckerman, 1995, Everitt et al., 2021] provide a representation that naturally fulfills these requirements. Similar to Influence Diagrams [Howard and Matheson, 1984], CIDs are commonly used to reason about decision-making tasks. CIDs further assume that the graph encodes the causal relationships between the nodes. We denote the set of parents of a node $X$ as $Pa_X$, the set of children as $Ch_X$, the set of ancestors as $Anc_X$, the set of descendants as $Desc_X$ and instantiations of random variables in lower-case. Each variable $X$ has a finite set of possible values $dom(X)$.

**Definition 1** (Causal influence diagram [Heckerman, 1995, Everitt et al., 2021]). A *causal influence diagram* (CID) is a Causal Bayesian Network $M = (G = (V, E), P)$, where $P$ is a joint probability distribution compatible with the conditional independences encoded in $G$. The nodes in $V$ are partitioned into decision ($D$), utility ($U$), and chance ($C$) nodes, $V = (D, U, C)$. Each utility node $U_i$ is associated with a real function $f_i$ of its parents $f_i : dom(Pa_{U_i}) \to \mathbb{R}$, which is referred to as the utility or loss function associated with $U_i$.

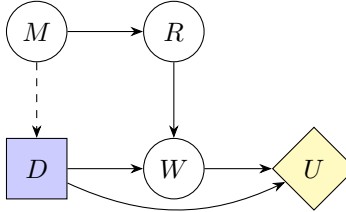

Figure 1: A CID representing a mediated decision task with $\mathbf{U} = \{U\}$, $\mathbf{D} = \{D\}$, and $\mathbf{C} = \{M, R, W\}$, where $M$ denotes the month of the year, $R$ indicates rainfall, and $W$ the lawn wetness. An AI agent controls a sprinkler ($D$) with the goal of keeping the grass wet ($W = 1$). Directed edges indicate causal relationships except the edge that enters $D$. The utility function $U$ depends on both wetness and sprinkler use, rewarding $W = 1$ and discouraging water wastage.

The chance nodes correspond to variables describing the environment. A decision task represented with a CID containing a decision node $D$ and a utility node $U$ is said to be *mediated* if $Desc_D \cap Anc_U \neq \emptyset$, that is, if a decision influences some part of the environment that is relevant to the task. For example, a doctor's treatment choice (decision $D$) alters a patient's condition (environment variable $C$), which affects recovery (utility $U$). Edges pointing into decision nodes are called *informational links*, they indicate which variables the agent observes when making a decision. All other edges are *causal links*, representing causal relationships. We prove that an agent capable of adapting to distribution shifts must have learned the CID of its environment by showing that it is possible to recover the causal structure and the Conditional Probability Tables (CPTs) of the variables describing the environment by observing the agent's optimal policies under distribution shifts. Note that this setup is unsuitable for traditional causal discovery algorithms like PC [Spirtes and Glymour, 1991] and FCI [Spirtes, 2001] because we do not have access to the joint probability distribution of the variables or any sample data. In this paper, the terms *adaptable agent* and *robust agent* refer to an agent that maintains optimal performance across all the possible distribution shifts in its environment.

Each agent may correspond to a different set of decision nodes and have access to a distinct subset of observable variables. When making a decision $D$, the agent only observes the variables that are parents of $D$. An example of a CID modeling the sprinkler story can be found in Figure 1.

**Modeling distribution shifts.** A key concept in this setting is domain dependence [Richens and Everitt, 2024], meaning that no single policy $\pi$ is optimal under all possible distribution shifts.

**Definition 2** (Domain dependence [Richens and Everitt, 2024]). *A CID $M$ is said to satisfy domain dependence if there exist $P(C = c)$ and $P'(C = c)$, both compatible with the CID $M$ such that $\pi^* = \arg\max_\pi \mathbb{E}_P^\pi[U] \implies \pi^* \neq \arg\max_\pi \mathbb{E}_{P'}^\pi[U]$.*

Domain dependence rules out trivial cases where the optimal policy remains the same under all distribution shifts. When it does not hold, any agent that is optimal in the unshifted environment would automatically be robust, making causal reasoning unnecessary for adaptation.

Following the work of Richens and Everitt [2024], we represent distribution shifts as mixtures of local interventions. Given a random variable $X$ with $x_1, \ldots, x_n$ as possible observable values, a local intervention on $X$ is a function that maps each observable value $x_i$ to a new observable value $f(x_i) \in \{x_1, \ldots, x_n\}$. In other words, local interventions deterministically reassign a random variable's outcomes independently of other variables.

**Definition 3** (Local intervention [Richens and Everitt, 2024]). *Local intervention $\sigma$ on $X$ involves applying a map to the states of $X$ that is not conditional on any other endogenous variables, $x \mapsto f(x)$. The conditional probability distribution of $X$ is modified as follows:*

$$P(x \mid \mathbf{pa}_X; \sigma) := \sum_{x': f(x')=x} P(x' \mid \mathbf{pa}_X) \tag{1}$$

We use the notation $\sigma = do(X = f(x))$ indicating that variable $X$ is assigned the value $f(x)$. Local interventions are a subset of soft interventions, with hard interventions as a special case.

A local intervention has limited capacity to model distribution shifts. For instance, it cannot model the shift from a coin that always lands on heads to a fair coin because a local intervention must deterministically map the observable value 'head' to another observable value. Therefore, we now report the concept of a mixture of interventions [Richens and Everitt, 2024]. This mixture is a convex combination $\sigma^* = \sum_i p_i \sigma_i$ of interventions $\sigma_i$, where each coefficient $p_i$ represents the probability that $\sigma_i$ is used to map the observable value of $X$.

**Definition 4** (Mixture of interventions [Richens and Everitt, 2024]). A *mixture of interventions* $\sigma^* = \sum_i p_i \sigma_i$ for $\sum_i p_i = 1$ performs intervention $\sigma_i$ with probability $p_i$. Formally:

$$P(x \mid pa_x; \sigma^*) = \sum_i p_i P(x \mid pa_x; \sigma_i) \tag{2}$$

We consider mixtures of local interventions on the environment variables, that is, on the chance nodes of the CID. An example of how a mixture of local interventions can represent a distribution shift is provided in Appendix D.

We use optimal policy oracles to formalize the agent's understanding of optimal behavior under distribution shifts. A policy for a decision $D$ is a set of conditional distributions on $dom(D)$ given each possible instantiation of its parent variables $Pa(D)$. Let $D$ be a decision variable taking values in $d \in dom(D)$. Given a set $\Sigma$ of interventions on the environment variables corresponding to chance nodes, an optimal policy oracle is a map that for any intervention (distribution shift) $\sigma \in \Sigma$ returns the corresponding optimal policy $\pi_\sigma(D \mid Pa_D)$.

**Definition 5** (Policy oracle [Richens and Everitt, 2024]). A policy oracle for a set of interventions $\Sigma$ is a map $\Pi_\Sigma : \sigma \mapsto \pi_\sigma(D \mid Pa_D) \; \forall \; \sigma \in \Sigma$. A policy oracle $\Pi_\Sigma^*$ is optimal if for each $\sigma \in \Sigma$, $\Pi_\Sigma^*(\sigma) = \pi_\sigma^*(D \mid Pa_D) = \arg\max_\pi \mathbb{E}[U \mid do(D = \pi(pa_D)); \sigma]$.

As a simple example, consider a game where an agent has to guess the outcome of a die roll $X$, the utility function is $U \coloneqq 1$ if $D = X$, and 0 otherwise. Let $\sigma$ be an intervention that corresponds to a distribution shift that makes the die always land on 3. Then, querying the policy oracle with $\sigma$ would return a policy $\pi_\sigma(D)$ so that $\pi_\sigma(D = 3) = 1$.

In most real-world settings, such an ideal agent may not exist. However, previous results have shown that for unmediated tasks, relaxing the optimality assumption for the policy oracle still requires the agent to have learned an approximate causal model [Richens and Everitt, 2024]. This suggests that the necessity to learn an exact or approximate causal model is a stable property that holds even for suboptimal agents.

In our work, we rely on Algorithm 1 from Richens and Everitt [2024], which takes as input a utility function $U$, an optimal policy oracle, an intervention $\sigma \in \Sigma$, and a parameter $N$ that controls the number of samples. For any local intervention $\sigma \in \Sigma$, let $d$ be the deterministic optimal decision under the shift induced by $\sigma$. By domain dependence, there exists a hard intervention $\sigma'$ such that $d$ is no longer optimal, with $d_2$ as the new optimal decision. Considering the mixture $\sigma(q) \coloneqq q\sigma + (1-q)\sigma'$, there exists a value $q_{crit}$ for $q$ such that $d_2$ and another decision $d_1$ are both optimal. The algorithm returns $q_{crit}$, $d_1$, and $d_2$. The value $q_{crit}$ is used to compute the CPTs of the chance variables which we use to reconstruct the causal graph. For an example see Appendix E.1 and Figure 7.

## 3 Learning Causal Influence Diagrams with Adaptable Agents

In Section 3.1, we present LearnCID, an algorithm for recovering the environment's CID from an adaptable agent, modeled as an optimal policy oracle. This algorithm establishes that acquiring causal knowledge is necessary for agents to be robust to distribution shifts, even in mediated decision tasks. In Section 3.2, we discuss the implications of our results for multi-agent environments. Finally, in Section 3.3 we outline the relation between CIDs and POMDPs and explain how our approach extends to sequential decision tasks modeled by POMDPs.

### 3.1 LearnCID algorithm for eliciting causal knowledge from adaptable agents

Under specific assumptions detailed below, the LearnCID algorithm (Algorithm 1) enables the reconstruction of the underlying causal model by identifying the CID structure and the CPTs for the variables corresponding to chance nodes that are not children of the decision node.

*Assumption pertaining to causal discovery*

**A1** The CID satisfies faithfulness [Spirtes et al., 1991] and causal sufficiency [Pearl, 2009]. Faithfulness means that every conditional independence in the joint probability $P$ of the CID also holds in the graph $G$. A set of variables in a CID is sufficient if it includes all common causes.

*Assumptions pertaining to the CID structure*

**A2** Given the CID $M = (G = (V, E), P)$, the partition of $V$ into $(D, U, C)$ is known.

**A3** The CID contains exactly one decision node $D$ and one utility node $U$ ($|D| = |U| = 1$).
For CIDs with multiple utility nodes, we can run the algorithm once for each utility node. Note that the optimal policy oracle depends on both the specific CID and decision node, so different utility node selections might correspond to different optimal policy oracles.

**A4** The set of parents of both $D$ and $Ch_D$, and the CPTs of the children of $D$ are known.
Motivations for Assumption A4 can be found in Appendix B.

**A5** $D$ is a parent of $U$.

**A6** All chance nodes are ancestors of $U$.
Chance nodes that are not ancestors of $U$ (and consequently, since $D \in Pa_U$, not ancestors of $D$) have no influence on the decision task. LearnCID would ignore these nodes, and their associated causal structure and CPTs would not be recovered.

*Assumptions pertaining to utility evaluation and decision policies*

**A7** The utility function $f$ associated with the utility node $U$ is fully specified.
The utility function's functional form is known, which tells us all the variables involved in calculating the utility. These variables appear in the causal graph as parents of the utility node.

**A8** We have access to a set $\Sigma$ of all possible mixtures of local interventions, along with the optimal policy oracle $\Pi^*_\Sigma$ for decision node $D$.

**A9** There exist no decision $d^* \in dom(D)$ that is optimal for all instantiations of $Pa_U \setminus D$.
Assumption A9 states that there does not exist a single decision $d \in Dom(D)$ that is optimal regardless of the value taken by the chance nodes that are parents of $U$. If Assumption A7 holds, Assumption A9 is testable by substituting different value combinations for the parent nodes of $U$. While Assumption A9 is equivalent to domain dependence (Definition 2) in unmediated decision tasks, this equivalence breaks down in the general mediated case. Nevertheless, Assumption A9 is sufficient to guarantee domain dependence.

The set of assumptions in Richens and Everitt [2024] is identical to that above, except we drop the unmediated task assumption. Under the no-mediation assumption, Assumption A4 corresponds to knowing the parents of $D$, A9 corresponds to domain dependence, and Assumption A4 together with Assumption A5 correspond to assuming the knowledge of the parents of $D$ and $U$. In the unmediated task case, Assumption A5 is implied by domain dependence [Richens and Everitt, 2024].

In this setting, Assumption A2 implies that the variables comprising the causal model are known. Beyond domain knowledge, there exist techniques to learn these variables; for instance, substantial work in causal representation learning aims, among other goals, to identify latent variables for causal models directly from data [Schölkopf et al., 2021, Varici et al., 2024].

The following theorem states that for a CID satisfying A9, domain dependence also holds.

**Theorem 1.** *Let $M = (G, P)$ be a CID where $Desc_D \cap Anc_U \neq \emptyset$. A9 $\implies$ Domain dependence.*

*Proof.* See Appendix C. $\qquad\square$

While under no mediation domain dependence implies A9 [Richens and Everitt, 2024], this no longer holds in the mediated case. Theorem 1 shows that A9 implies domain dependence, making the two equivalent in the unmediated setting. This is further discussed in Appendix C.

### 3.1.1 Algorithm formulation

The LearnCID algorithm reconstructs the structure and CPTs of the CID by iteratively exploring chance nodes connected to the utility node $U$. For each chance node $X$ with a known path to $U$, we first identify a set of candidate parent chance nodes $Pa^*_X$, also making use of prior knowledge: $V_{kwn}$ the subset of chance nodes whose parents are fully known, and $G'$ a graph containing known edges. Consider the following local intervention:

$$f_x(X) \leftarrow \begin{cases} x, & \text{if } X = x \\ x', & \text{otherwise} \end{cases} \tag{3}$$

---
**Algorithm 1** LearnCID
---
**Input:** Nodes $V = \{\{D\}, \{U\}, C\}$, graph $G'$ on $V$ containing known edges, set of chance nodes with all known parents $V_{\text{kwn}}$, set of known CPTs $\Theta'$, and the number of samples $N$ to estimate $q_{\text{crit}}$.
**Output:** The CID's graph $G$, and the set of CPTs $\Theta$,

1: Mark all chance nodes as unvisited.
2: **while** there are still unvisited chance nodes with a known path to $U$, starting from the parents of $U$ not children of $D$ **do**
3:     Let $X$ be an unvisited chance node with a known path to $U$.
4:     Let $I_p$ be the set of intermediate chance nodes on a directed path $p$ from $X$ to $U$, or to $D$ if no such path to $U$ exists.
5:     Let $C_X$ be the set of chance nodes not in $I_p$
6:     **if** $X \in V_{\text{kwn}}$ **then** $Pa_X^* \leftarrow Pa_X$ **else** $Pa_X^* \leftarrow C_X \backslash Desc_X$
7:     **for** each instantiation $x$ of $X$ **do**
8:         **for** each $Pa_{X,i}^* \in Pa_X^*$ **do**
9:             **for** each instantiation of variables $c_1$ in $C_X \setminus \left( \{Pa_{X,i}^*\} \cup Desc_X \right)$ **do**
10:                 Let $c_2$ be a valid instantiation of the variables in $C_X \cap Desc_X$
11:                 Construct $c$ instantiation of $C_X \setminus Pa_{X,i}^*$ by combining $c_1$ and $c_2$
12:                 Initialize $\sigma_{Pa_{X,i}^*}(c)$ as in Equation 4.
13:                 **for** each $\sigma \in \sigma_{Pa_{X,i}^*}$ **do**
14:                     Estimate $q_{\text{crit}}, d_1, d_2, pa_U'$ using $\text{ALG}q_{\text{crit}}(U, \Pi_{\Sigma}^*, N, \sigma)$.
15:                     Compute $P(X = x \mid pa_x; \sigma)$ using $\Theta$ and Equations 5 and 6.
16:                 **if** $Pa_{X,i}^* \to X \notin G'$ and $\exists \sigma, \sigma' \in \sigma_{Pa_{X,i}^*}$ s.t. $P(x \mid pa_X; \sigma) \neq P(x \mid pa_X; \sigma')$ **then**
17:                     Add $Pa_{X,i}^* \to X$ to $G'$
18:         **for** each $pa_X \in dom(Pa_X)$ **do**
19:             $P(x \mid pa_X) \leftarrow P(x \mid pa_X; \sigma)$ for any hard intervention $\sigma$ compatible with $pa_X$.
20: **return** the updated CID's graph $G'$, and the set of CPTs $\Theta'$.
---

where $x'$ is an arbitrary observable value for $X$ different from $x$. Let $C_X$ represent the set of all chance nodes except $X$ and the intermediate nodes on a directed path from $X$ to either $U$ or $D$. For example, in a CID $X_4 \to X_3 \to X_2 \to X_1 \to U \leftarrow D$, then $C_{X_3} = \{X_4\}$ because $X_2 \to X_1$ is on the only directed path between $X_3$ and $U$. For each candidate parent $Pa_{X,i}^* \in Pa_X^*$, we define a family of interventions $\sigma_{Pa_{X,i}^*}$.

$$\sigma_{Pa_{X,i}^*}(c) \leftarrow \{do(Pa_{X,i}^* = pa_{X,i}^*, C_X \setminus \{Pa_{X,i}^*\} = c, X = f_x(X)) \mid pa_{X,i}^* \in dom(Pa_{X,i}^*)\} \quad (4)$$

Note that in these interventions, all variables to which we assign a specific value are fixed except one, $Pa_{X,i}^*$, which is varied across different interventions. To compute the CPTs, we use $\text{ALG}q_{\text{crit}}$ (Algorithm 1 in Richens and Everitt [2024]). Given an intervention $\sigma_1$ and the corresponding optimal decision $d_1$ obtained from the policy oracle, the algorithm finds another intervention (a distribution shift) $\sigma_2$ under which $d_1$ is no longer optimal. The existence of such an intervention is guaranteed by A9. It then defines a mixture of interventions $\sigma(q) := q\sigma_1 + (1-q)\sigma_2$ and identifies $q_{\text{crit}}$, the convex coefficient at which $d_1$ performs as well as an optimal decision $d_2$ under $\sigma_2$. In other words, at $q_{\text{crit}}$ the expected utilities of $d_1$ and $d_2$ under $\sigma(q_{\text{crit}})$ are equal. Rearranging this equality leads to Equations 5 and 6, which use the value of $q_{\text{crit}}$ to compute the interventional distributions.

Let $C_1, \ldots, C_k$ be the intermediate chance nodes in a directed path from $X$ to $U$ or $D$. If $C_1 \in Pa_U$ let $\mathcal{C} := \{C_1, \ldots, C_k\}$ otherwise let $\mathcal{C} := \{C_2, \ldots, C_k\}$. For both $x$ and $x'$, we compute:

$$\beta(x) := \sum_{c \in dom(\mathcal{C})} \prod_{i=1}^{k} P(c_i \mid pa_{C_i})[U(d_2, c) - U(d_1, c)] \quad (5)$$

Observe that the right-hand side of Equation 5 depends on $X$ for determining the set containing the path of chance nodes $\mathcal{C}$, and depends on the specific instantiation $x$ of $X$ because $X$ is the parent of either a chance node in $\mathcal{C}$, the decision node $D$, or the utility node $U$. Let $pa_U'$ be the instantiation of the variables associated with the chance nodes that are parents of $U$ under $\sigma_2$. Using Equation 5, we

can compute $P(x \mid pa_X; \sigma)$ as:

$$P(x \mid pa_X; \sigma) = \frac{(1 - \frac{1}{q_{crit}})[U(d_2, pa_U') - U(d_1, pa_U')] - \beta(x')}{\beta(x) - \beta(x')} \qquad (6)$$

We proceed by testing whether any pair of interventions produces distinct interventional conditional distributions for $X$. If so, the way the interventions in $\sigma_{Pa_{X,i}^*}$ are defined ensures that $Pa_{X,i}^*$ is a parent of $X$. We can then recover the observational conditional distribution of $X$ from the interventional ones by noting that, according to Equation 1, for any $\sigma \in \sigma_{Pa_{X,i}^*}$ that sets $Pa_X$ to $pa_X$, it holds that $P(x \mid pa_X; \sigma) = P(x \mid pa_X)$. This process is repeated until all chance nodes that are ancestors of $U$ have been processed, returning the learned structure and CPTs.

An example of an application of Algorithm 1 (LearnCID) to a single-agent CID is provided in Appendix E.1. The proof of correctness for Algorithm 1 can be found in Appendix A. The algorithmic complexity of Algorithm 1 is discussed in Appendix A.3.

With Theorem 2, we prove that **agents robust to distribution shifts must have learned a causal model of their environment even under mediation** and formalize the correctness of Algorithm 1:

**Theorem 2.** *Let $M$ be a CID satisfying Assumptions A1~A9. Suppose for each $C_i$, we know a set $\widehat{Pa}_{C_i} \subseteq Pa_{C_i}$ and a set of nodes $V_{kwn} \subseteq V$ where $C_i \in V_{kwn} \iff \widehat{Pa}_{C_i} = Pa_{C_i}$. Let $\{\pi_\sigma^*(d|pa_D)\}_{\sigma \in \Sigma}$ denote the set of optimal policies available where $\pi_\sigma^*(d|pa_D)$ is an optimal policy under the intervention $\sigma$ and $\Sigma$ is the set of all mixtures of local interventions.*
*Then, for almost all CIDs, the graph $G$ and the joint distribution $P$ over all the ancestors of the utility node $Anc_U$ can be identified.*

*Proof.* See Appendix A. □

In Richens and Everitt [2024], it is claimed that the set of optimal policies is *almost always* sufficient to identify all causal relationships, where *almost always* means except on a set of parameters of Lebesgue measure zero. While this holds for unmediated tasks, we show that in the mediated case, the children of the decision node $D$ cannot, in general, be identified, which in turn motivates Assumption A4.

**Theorem 3.** *Let $M = (G = \{V, E\}, P)$ be a single decision/single utility CID, assume we know $D, U, C$, and $Pa(U), Pa(D)$. Let $\Sigma$ be the set of all mixtures of local interventions, $\Pi_\Sigma^*$ be an optimal policy oracle for $D$. Then in general $Ch_D$ cannot be uniquely determined.*

*Proof.* See Appendix B. □

### 3.2 Implications for multi-agent environments

In this section we show how to apply our results to multi-agent systems represented by multi-decision CIDs. The fact that an adaptable agent must necessarily learn a causal model of its environment has significant implications when extending this concept to multi-agent systems. Under the assumptions of Algorithm 1, the causal model of the nodes in $Anc_U$, the variables influencing the utility function, can be recovered. Now suppose we introduce a new agent $A'$ into the same environment, by adding its decision nodes and related edges. The existing causal knowledge over $U \cup Anc_U$ can, in principle, inform $A'$ and aid in designing policies that make it adaptable to distribution shifts [Bareinboim et al., 2024]. Let $\mathbf{U}'$ denote the set of utility functions that $A'$ tries to optimize. Intuitively, the extent to which the original agent's causal knowledge can be transferred to $A'$ depends on how much of the environment they share, more precisely, on the intersection $Anc_U \cap \left(\bigcup_{U' \in \mathbf{U}'} Anc_{U'}\right)$. We can expect this transferability to be maximal when $Anc_U = \bigcup_{U' \in \mathbf{U}'} Anc_{U'}$ and minimal when $Anc_U \cap \left(\bigcup_{U' \in \mathbf{U}'} Anc_{U'}\right) = \emptyset$. As an example, Consider a factory with two agents: agent A, the production manager, determines how much to produce to maximize profit, while agent $B$, the maintenance manager, checks the machine conditions and schedules maintenance actions in order to avoid breakdowns and maximize uptime. In this example $Anc_{U_A} \cap Anc_{U_B}$ is not empty, since a breakdown reduces uptime and could prevent agent A from producing the desired quantity of goods. Agent A's causal knowledge about breakdowns can be informative for agent B and vice versa.

**Learning multi-decision CIDs.** Even if LearnCID can be applied only to single-decision CIDs, there exist different approaches to handle multi-decision CIDs. We propose two approaches for when

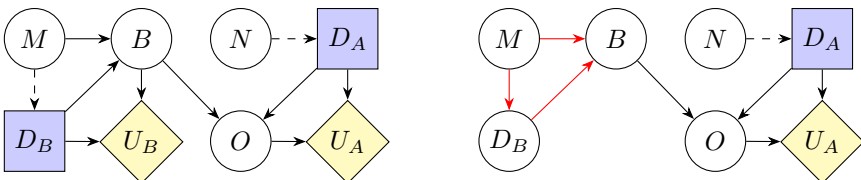

Figure 2: On the left, a CID representing a factory decision-making scenario with $\mathbf{U} = \{U_A, U_B\}$, $\mathbf{D} = \{D_A, D_B\}$, and $\mathbf{C} = \{M, B, O, N\}$. Here, $M$ denotes machine condition, $B$ machine breakdown, $O$ the production output, and $N$ the product demand. Agent A (the production manager) controls $D_A$, adjusting the production rate in response to $N$, while Agent B (the maintenance manager) controls $D_B$, scheduling maintenance actions informed by $M$. Utilities $U_A$ and $U_B$ reflect production profit and machine uptime, respectively. On the right, the corresponding transformed CID used to apply LearnCID, with $D_A$ as the primary decision node and $D_B$ as a secondary decision node, assuming $D_B$ is assigned a faithful policy. Edges marked in red denote unknown dependencies.

all agents are able to influence the same utility function and where at least one agent optimizes it. In both approaches the optimizing agent may correspond to multiple decision nodes. We designate one of these as the primary decision node and require the single-decision assumptions to hold for it (knowledge of its children, their CPTs, and its parents). All remaining decision nodes are referred to as secondary decision nodes. In the example described in Figure 2, assuming agent A is robust, its decision node $D_A$ is designated as the primary decision node and $D_B$ as secondary decision node. If the multi-decision CID contains multiple utility nodes, we can prune all but one and then run LearnCID to recover its ancestors and their CPTs. It is then possible to repeat the procedure for other utility nodes in the environment, provided that for each such node there is at least one optimal policy oracle available corresponding to a decision node that optimizes the associated utility function and that the other LearnCID assumptions are satisfied.

1. Let $\{D_i\}_i$ be the set of all secondary decision nodes. A possible approach consists in assuming that the decision associated to each secondary decision node $D_i$ is determined by a faithful policy $\pi_i$, i.e. a policy that actually depends on $Pa_{D_i}$. Since the policy is fixed, we can convert these decision nodes to chance nodes with the corresponding, potentially unknown, policy $\pi_i$ as CPT. If we assume the availability of an optimal policy oracle for the primary decision node, we can apply Algorithm 1 to recover the full CID.

2. As a special case, if we know the parent sets of all secondary decision nodes $\{D_i\}_i$. It is then possible for us to assign any faithful policy to each decision node $D_i$. Then, we can again convert these nodes to chance nodes by using their policies as CPTs. Assuming the existence of an optimal policy oracle for the primary decision node, we can apply Algorithm 1 to learn the full CID.

Once the CPTs for all chance nodes are learned, it becomes possible to determine an optimal policy for every decision node in the original graph under any distribution shift [Shachter, 1986, Koller and Milch, 2001, Hammond et al., 2021, Bareinboim et al., 2022]. In Appendix E.2, we provide an example of an application of LearnCID to a system where two agents cooperate.

### 3.3 Applying LearnCID to Partially Observable Markov Decision Processes

Let $M = (S, \mathcal{A}, T, R, \Omega, O, \gamma)$ be a Partially Observable Markov Decision Process (POMDP) [Kaelbling et al., 1998] where $S$ is the set of states, $A$ the set of actions/decisions, $T : S \times A \rightarrow \Pi(S)$ is the state-transition function where $\Pi(S)$ is the set of probability distributions over the set of states, $R : S \times A \rightarrow \mathbb{R}$ is a reward function, $\Omega$ is the set of observables (values of observable variables), $O$ is the set of conditional observation probabilities, and $\gamma \in [0, 1)$ is the discount factor.

POMDPs form a class of models for sequential decision-making that typically involve mediation, as state transitions generally depend on the agent's actions. In this section, we demonstrate that under mild assumptions, given a POMDP with unknown state-transition function $T$ and states described by a finite set of discrete variables, a POMDP can be represented with a CID with a temporal component. Given access to an optimal policy oracle, Algorithm 1 can then be used to learn both the intra- and inter-temporal causal structure, as well as the CPTs and the state-transition function.

**Structural constraints.** We make the following assumptions on the POMDP:

P1 Time-homogeneity, i.e. the transition probabilities do not change over time.

P2 The set of observable variables is fixed and the agent gets an observation from these variables at every timestep.
Formally, each state $s$ in $S$ is described by a finite set of variables $V$ (i.e., there exists a bijective function $f : dom(V) \rightarrow S$), there exists a subset of observable variables $\mathbf{V_o} \subseteq \mathbf{V}$, and for every state $s$ in $S$ and every action $a$ in $A$ it holds that $O(s, a, v_o) = 1$, where $v_o$ is the instantiation of the observable variables at the current state.

Note that under Assumptions P1 and P2, we can model the state space of a POMDP using a CID with $Pa_D \coloneqq \mathbf{V}_o$. If the set of observable variables coincides with the set of all state variables then we get a Markov Decision Process (MDP) as a special case of a POMDP.

**Causal representation of POMDPs.** By identifying the observable variables of the POMDP as the parents of decision nodes, the reward function as the utility function associated with a utility node, and all other unobserved variables as chance nodes that are not parents of decision nodes, we can observe that a POMDP can be seen as a CID unrolled over time. Previous work has explored analogous causal representations of POMDPs [Everitt et al., 2019, Bareinboim et al., 2024].

The mapping from a POMDP to a CID with unknown structure is described in Algorithm 2. Since we want to apply Algorithm 1 to this CID, we further need to assume each variable is discrete, the POMDP reward directly depends on the action (equivalent to $D \in Pa_U$). In this case, we have a slightly weaker condition for the Markov blanket of the decision node: other than the parents of $D$, we only need to know which chance nodes are children of $D$ in the same timestep, their CPTs, and their parents, while we can ignore the chance nodes that are children of $D$ in the following timesteps (e.g., considering the POMDP described in Figure 3, we do not need to know that $X_{t+1}$ is a child of $D_t$ because they belong to different timesteps). Due to the Markov property of POMDPs, direct causal relationships can exist only within a single timestep or, when inter-temporal, from past to future between consecutive timesteps. Note that the causal structure, the CPTs, and the state-transition function associated with the POMDP are independent of the discount factor. Therefore, even if the CID obtained with Algorithm 2 does not contain any information about the original discount factor, as long as the assumptions of LearnCID are satisfied, we can still recover these components.

---

**Algorithm 2** POMDPtoCID

---

**Input:** POMDP $(S, \mathcal{A}, T, R, \Omega, O, \gamma)$ with unknown state-transition function $T$, states described by a finite set of variables $\mathbf{V}$, and the subset of observable variables $\mathbf{V}_o \subseteq \mathbf{V}$.
**Output:** Corresponding CID's graph $G$ without causal arcs involving chance nodes.

1: Initialize $G$ as an empty graph.
2: Add decision node $D_t$ to CID's graph $G$ with $\mathcal{A}$ as set of decisions.
3: Add chance node $D_{t-1}$ to CID's graph $G$.
4: **for** each variable $V \in \mathbf{V}$ **do**
5:     Add chance nodes corresponding to random variables $V_t$ and $V_{t-1}$ to CID's graph $G$.
6:     **if** $V \in \mathbf{V_o}$ **then**
7:         Add edge $V_t \rightarrow D_t$ and $V_{t-1} \rightarrow D_{t-1}$ to CID's graph $G$.
8: Define $\mathcal{U} : dom(V) \times dom(D) \rightarrow \mathbb{R}$ as $\mathcal{U}(v, d) \mapsto \mathcal{R}(f(v), d)$ for all $d \in dom(D)$ and $v \in dom(V)$
9: Add utility node $U_t$ with utility function $\mathcal{U}$ to $G$.
10: Add edge $D_t \rightarrow U_t$ with utility function $\mathcal{U}$ to $G$.
11: **return** CID's graph $G$.

---

**Learning intra- and inter-temporal causal relationships.** We can run LearnCID on the CID produced by Algorithm 2 to learn the causal relationships for variables both at the same time step and at different ones. The algorithm maps the variables, decisions, and rewards related to two consecutive timesteps to nodes of a CID. We now refer to these timesteps as $t-1$ and $t$. Therefore, for each variable $X$ partially describing the state of the POMDP, this newly defined CID contains two variables $X_{t-1}$ and $X_t$. Similarly, there will be two decision nodes $D_{t-1}$ and $D_t$ and two utility nodes $U_{t-1}$ and $U_t$. Observe that we can prune $U_{t-1}$, and, similarly to what we proposed for multi-agent settings,

Figure 3: Example of an application of Algorithm 2 and Algorithm 1 (LearnCID) to a time-homogeneous POMDPs. On the left, a causal representation of a POMDP. On the right, the corresponding CID with unknown edges marked in red. To apply LearnCID, we consider two time steps $t-1$ and $t$, with decision node $D_t$ and utility node $U_t$. We select any faithful policy for $D_{t-1}$, convert it into a chance node, and prune $U_{t-1}$. Time-homogeneity ensures that inter-temporal causal relationships are preserved across all time steps, allowing us to recover the full causal graph.

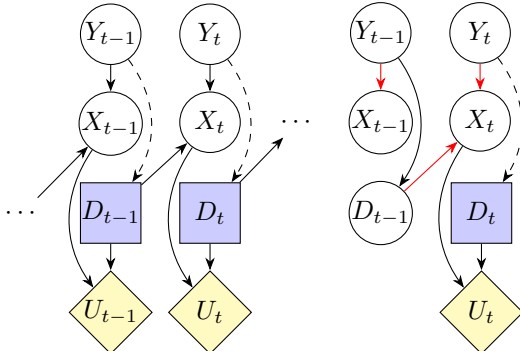

we convert $D_{t-1}$ to a chance node and assign any faithful policy as its CPT. This is possible because the chance nodes that are parents of decision nodes correspond to the observable variables which are known. Then, if the assumptions of LearnCID are satisfied, including having an optimal policy oracle available, then we can use it to learn all the missing causal relationships. For time-homogeneous POMDP, like in the example illustrated in Figure 3, this is sufficient to learn all the intra- and inter-temporal causal relationships of variables in $Anc_{U_t}$ for any timestep since they do not change when varying $t$ because this would imply a change in the transition probability which contradicts the time-homogeneity assumption.

**Learning the state-transition function.** Once we have learned the CPTs for all the chance nodes and we have fixed a policy $\pi$ for the agent, it is possible to recover the state-transition function $T$, i.e., for all states $s \in S$ and actions $a \in A$ we can find a probability distribution over the state of the process at the following timestep. Let $V_1^{(t)}, \ldots, V_n^{(t)}$ be the variables $\mathbf{V}$ associated with the chance nodes at timestep $t$ we observe that:

$$T(s^{(t-1)}, a^{(t-1)}) = P(s^{(t)}|s^{(t-1)}, a^{(t-1)}) = P(V_1^{(t)}, \ldots, V_n^{(t)}|v_1^{(t-1)}, \ldots, v_n^{(t-1)}, a^{(t-1)})$$

$$= \prod_{i|V_i \in Ch_D} \sum_{d \in \mathcal{A}} P(V_i^{(t)}|pa_{V_i}, d)\pi(d|pa_D) \prod_{i|V_i \in \mathbf{V} \setminus Ch_D} P(V_i^{(t)}|pa_{V_i}) \tag{7}$$

Since all the CPTs are known and the policy is fixed, we can compute the state-transition function for all states and actions.

**Non time-homogeneous case.** For non time-homogeneous POMDPs, this method only recovers causal links between two adjacent timesteps, as time-varying dynamics can change variable interactions. For example, irrigation may affect soil humidity in the dry season but not during the rainy season, so applying the algorithm on two rainy days could falsely suggest no causal link. One fix is to augment the state space to enforce time homogeneity [Puterman, 2005]. Alternatively, the method can be applied separately at each timestep, which is practical when time dependence is finite or periodic. Otherwise, it is possible to map each interval to a CID, prune intermediate utility nodes, and use our multi-decision CID approach. Each strategy has different oracle requirements: the augmented model needs time interventions, while the others require one oracle per time interval.

## 4 Conclusions and Future Work

In this work, we addressed the challenge of understanding the relationship between robustness to distribution shifts and an agent's causal understanding of its environment. While previous work established that robust agents encode the causal model in single-agent, unmediated tasks, we demonstrated that this connection also holds in mediated, multi-agent, and sequential settings. We presented an algorithm to show that it is possible to elicit the learned causal model from robust agents. In multi-agent systems, we showed how a single robust agent enables the discovery of the complete causal model, and how this could be used to learn optimal policies for other agents in the same environment. We further applied our approach to POMDPs, demonstrating that robust agents necessarily learn intra- and inter-temporal causal relationships and the state-transition function. These findings contribute to a theoretical foundation for world modeling approaches based on the extraction and combination of causal knowledge from robust agents and facilitate the exploration of approximate settings, where the agent's optimality assumption is relaxed.

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

# A  Correctness Proof for LearnCID

## A.1  Setup and Assumptions

Our goal is to show that agents capable of adapting to distribution shifts must have learned a causal model of their environment. To do this we show it is possible to recover the causal structure and the Conditional Probability Tables (CPTs) of the variables describing the environment in which the agents operate. This environment consists of both observable variables and hidden latent variables. We define a set of interventions modeling distribution shifts, and to learn the causal graph, we query an optimal policy oracle associated with one agent to get the optimal policy for that agent under the specified distribution shift. Observe that this setup is unsuitable for traditional causal discovery algorithms like PC [Spirtes and Glymour, 1991] and FCI [Spirtes, 2001] because we do not have access to the joint probability distribution of the variables or any sample data.

To model the causal relationships in the environment, we use Causal Influence Diagrams (CIDs) [Heckerman, 1995, Everitt et al., 2021]. Similar to Influence Diagrams [Howard and Matheson, 1984], CIDs are commonly used to reason about decision-making tasks. CIDs further assume that the graph encodes the causal relationships between the nodes. We denote the set of parents of a node $X$ as $Pa_X$, the set of children as $Ch_X$, the set of ancestors as $Anc_X$, the set of descendants as $Desc_X$ and instantiations of random variables in lower-case.

**Definition 1** (Causal influence diagram [Heckerman, 1995, Everitt et al., 2021]). A *causal influence diagram* (CID) is a Causal Bayesian Network $M = (G = \{V, E\}, P)$, where $P$ is a joint probability distribution compatible with the conditional independences encoded in $G$. The variables in $V$ are partitioned into decision, utility, and chance variables, $V = (D, U, C)$. Each utility node $U_i$ is associated with a real function $f_i$ of its parents $f_i : dom(Pa_{U_i}) \to \mathbb{R}$.

Each agent may correspond to a different set of decision nodes and have access to a distinct subset of observable variables. A variable observed by one agent may be latent for another. Additionally, considering a situation where an agent takes more than one decision, the set of variables that it observes when it takes one decision can differ from the one that it observes when taking another decision.

Following the work of [Richens and Everitt, 2024], we represent distribution shifts as mixtures of local interventions. Given a random variable $X$ with $x_1, \dots, x_n$ as possible observable values, a local intervention on $X$ is a function $\sigma : x_i \mapsto f(x_i)$ that maps each observable value $x_i$ to a new observable value $f(x_i)$. In other words, local interventions deterministically reassign a random variable's outcomes independently of other variables.

**Definition 3** (Local intervention [Richens and Everitt, 2024]). *Local intervention* $\sigma$ on $X$ involves applying a map to the states of $X$ that is not conditional on any other endogenous variables, $x \mapsto f(x)$. We use the notation $\sigma = do(X = f(x))$ (variable $X$ is assigned the state $f(x)$). Formally, this is a soft intervention on $X$ that transforms the conditional probability distribution as,

$$P(x \mid \mathbf{pa}_X; \sigma) := \sum_{x' : f(x') = x} P(x' \mid \mathbf{pa}_X) \tag{8}$$

In general, a local intervention has limited capacity to model distribution shifts. For instance, it cannot model the shift from a coin that always lands on heads to a fair coin because a local intervention must deterministically map the observable value 'head' to another observable value. Therefore, we now report the concept of a mixture of local interventions [Richens and Everitt, 2024]. This mixture is a convex combination $\sigma^* = \sum_i p_i \sigma_i$ of local interventions $\sigma_i$, where each coefficient $p_i$ represents the probability that $\sigma_i$ is used to map the observable value for $X$.

**Definition 4** (Mixture of interventions [Richens and Everitt, 2024]). A *mixture of interventions* $\sigma^* = \sum_i p_i \sigma_i$ for $\sum_i p_i = 1$ performs intervention $\sigma_i$ with probability $p_i$. Formally, $P(x \mid \sigma^*) = \sum_i p_i P(x \mid \sigma_i)$.

We use optimal policy oracles to formalize the agent's understanding of optimal behavior under distribution shifts. Let $D$ be a decision variable with observable values $d \in dom(D)$, given a set of interventions $\Sigma$, an optimal policy oracle is a map $\Pi_\Sigma^* : \sigma \mapsto \pi_\sigma(d \mid pa_D)$ for $\sigma \in \Sigma$, where $\pi_\sigma(d \mid pa_D)$ is the optimal policy under the distribution shift induced by the intervention $\sigma$.

**Definition 5** (Policy oracle [Richens and Everitt, 2024]). A policy oracle for a set of interventions $\Sigma$ is a map $\Pi_\Sigma : \sigma \mapsto \pi_\sigma(D \mid Pa_D) \; \forall \; \sigma \in \Sigma$. A policy oracle $\Pi_\Sigma^*$ is optimal if for each $\sigma \in \Sigma$, $\Pi_\Sigma^*(\sigma) = \pi_\sigma^*(D \mid Pa_D) = \arg\max_\pi \mathbb{E}[U \mid do(D = \pi(pa_D)); \sigma]$.

In our work, we rely on Algorithm 1 from Richens and Everitt [2024], which takes as input a utility function $U$, an optimal policy oracle, an intervention $\sigma \in \Sigma$, and a parameter $N$ that controls the number of samples. For any local intervention $\sigma \in \Sigma$, let $d$ be the deterministic optimal decision under the shift induced by $\sigma$. By Assumption A9, there exists a hard intervention $\sigma'$ such that $d$ is no longer optimal. Let $d_2$ be the deterministic optimal decision under $\sigma'$. Considering the mixture $\sigma(q) := q\sigma + (1 - q)\sigma'$, there exist a value $q_{crit}$ for $q$ such that $d_2$ and another decision $d_1$ are both optimal. The algorithm returns $q_{crit}$, $d_1$, and $d_2$.

Now, we list and motivate our assumptions.

*Assumption pertaining to causal discovery*

**Assumption 1.** The CID is faithful [Spirtes et al., 1991] and sufficient [Pearl, 2009].

Faithfulness implies that every conditional independence in the joint probability $P$ of the CID also holds in the graph $G$. A set of variables in a causal model is sufficient when it includes all common causes.

*Assumptions pertaining to the CID structure*

**Assumption 2.** Given the CID $M = (G = \{V, E\}, P)$ with $V = (D, U, C)$, the set of nodes and the partition $(D, U, C)$ is known.

The set of nodes together with the node partition $(D, U, C)$ is known, therefore we know all variables in the system and the type of each node (decision, utility, or chance).

**Assumption 3.** The CID contains exactly one decision node $D$ and one utility node $U$.

Despite assumption 3, this algorithm can be applied to multi-decision CIDs. It is possible to find further details in the main paper.

**Assumption 4.** The Markov blanket of decision node $D$ is known. We also know all the edges between these nodes. The CPTs of chance nodes that are children of $D$ are known.

Motivations for Assumption 4 can be found in Appendix B.

**Assumption 5.** $D$ is a parent of $U$.

We assume $D$ is a parent of $U$. In particular, this plays a role in the proof of Lemma 1.

**Assumption 6.** All chance nodes are ancestors of $U$.

In the presence of chance nodes that are neither ancestors of $D$ or $U$, these nodes do not have any influence on the decision task. LearnCID would simply not process those nodes and the related causal structure or CPT would not be recovered.

*Assumptions pertaining to utility evaluation and decision policies*

**Assumption 7.** The utility function $f$ associated with the utility node $U$ is fully specified.

The utility function's functional form is known, which tells us all the variables involved in calculating the utility. These variables appear in the causal graph as parents of the utility node.

**Assumption 8.** We have access to a set $\Sigma$ of all possible mixtures of local interventions, along with the optimal policy oracle $\Pi_\Sigma^*$ for decision node $D$.

**Assumption 9.** There exists no $d^* \in dom(D)$ such that $d^* \in \arg\max_d U(d, x) \; \forall \, x \in dom(Pa_U \backslash \{D\})$.

Assumption 9 states that there does not exist a single decision $d \in Dom(D)$ that is optimal regardless of the value taken by the chance nodes that are parents of $U$. When Assumption 7 holds, it is possible to verify if the CID we are considering satisfies Assumption 9 by computing the utility for different instantiations of variables associated with the parents of $U$.

Observe that under Assumptions 3, 5 and 9, there must be at least one chance node that is a parent of $U$, otherwise the utility function would only depend on the decision and therefore there would exist at least one optimal decision that would violate Assumption 9. We provide a discussion the Assumption 9 and domain dependence in Appendix C.

## A.2 Proof

**Lemma 1.** *Under Assumptions 1,3,5 and 7, given a CID $M = (G, P)$, for any given local intervention $\sigma$ there is a single deterministic optimal policy for almost all P, U.*

*Proof.* When there is no other directed path from $D$ to $U$ except $D \to U$, i.e. $Desc_D \cap Anc_U = \emptyset$ (unmediated case) the statement is proven by Lemma 3 in Richens and Everitt [2024]. Assume there exist a path from $D$ to $U$ other than $D \to U$, i.e. $Desc_D \cap Anc_U \neq \emptyset$. Let $\mathbf{Z} = Anc_U \setminus Pa_D$, and $\mathbf{X} = Pa_U \setminus \{D\}$ Assume $\exists d_1, d_2$ s.t. both are optimal decisions in the context $pa_D$, we divide the proof in two cases.

**Case 1**. Assume $Ch_D \cap C \neq C$.

Since $d_1$ and $d_2$ are both optimal we have:

$$\mathbb{E}[u \mid pa_D, do(D = d_1); \sigma] = \mathbb{E}[u \mid pa_D, do(D = d_2); \sigma] \tag{9}$$

Since $D \perp\!\!\!\perp U \mid Pa_D$ in $G_{\underline{D}}$, then according to rule 2 of do-calculus [Pearl, 2009] we can rewrite the equation as:

$$\mathbb{E}[u \mid pa_D, D = d_1; \sigma] = \mathbb{E}[u \mid pa_D, D = d_2; \sigma] \tag{10}$$

Equivalently:

$$\sum_z U(d_1, x) P(z \mid pa_D, D = d_1; \sigma) = \sum_z U(d_2, x) P(z \mid pa_D, D = d_2; \sigma) \tag{11}$$

$$\sum_z U(d_1, x) P(z \mid pa_D, D = d_1; \sigma) - \sum_z U(d_2, x) P(z \mid pa_D, D = d_2; \sigma) = 0 \tag{12}$$

Let us define $S_1 := \mathbf{Z} \setminus Ch_D$ and $\overline{Pa}_{C_i} := Pa_{C_i} \setminus \{D\}$. We factorize the joint distribution:

$$P(z \mid pa_D, D = d_1; \sigma) = \prod_{Z_i \in S_1} P(Z_i \mid pa_{Z_i}; \sigma) \prod_{C_i \in Ch_D} P(C_i \mid \overline{pa}_{C_i}, D = d_1; \sigma) \tag{13}$$

Same for $d_2$. Now, we can rewrite Equation 12 as:

$$\sum_z \prod_{Z_i \in S_1} P(Z_i \mid pa_{Z_i}; \sigma) \Big[ U(d_1, x) \prod_{C_i \in Ch_D} P(C_i \mid \overline{pa}_{C_i}, D = d_1; \sigma) \\ - U(d_2, x) \prod_{C_i \in Ch_D} P(C_i \mid \overline{pa}_{C_i}, D = d_2; \sigma) \Big] = 0 \tag{14}$$

We can define $a(d, z) := U(d, x) \prod_{C_i \in Ch_D} P(C_i \mid \overline{pa}_{C_i}, D = d; \sigma)$.

Observe that:

$$\sum_z \prod_{Z_i \in S_1} P(Z_i \mid pa_{Z_i}; \sigma) \big[ a(d_1, z) - a(d_2, z) \big] = 0 \tag{15}$$

Is a polynomial equation with variables $\forall i.\ P(Z_i \mid pa_{Z_i}; \sigma)$. Also notice that if we rewrite each of these variables using Definition 3 (Local intervention) $P(Z_i \mid pa_{Z_i}; \sigma) = \sum_{z_i' : f(z_i') = z_i} P(z_i' \mid pa_{Z_i})$ the equation is still a polynomial equation with all the parameters of the CPTs, excluding those related to the children of $D$, as variables. If the polynomial is not trivial ($\exists d, z.\ a(d_1, z) - a(d_2, z) \neq 0$) then the Lebesgue measure of the solution of this equation is zero [Okamoto, 1973], and since the set of parameters that allow for multiple optimal solutions has measure zero along at least one dimension, it follows that the whole set has measure zero.

Now, let us consider when the polynomial is trivial. Then for all $d, z$ we have:

$$a(d_1, z) - a(d_2, z) = 0 \tag{16}$$

$$U(d_1, x) \prod_{C_i \in Ch_D} P(C_i | \overline{pa}_{C_i}, D = d_1; \sigma) - U(d_2, x) \prod_{C_i \in Ch_D} P(C_i | \overline{pa}_{C_i}, D = d_2; \sigma) = 0 \tag{17}$$

Again, this is a finite number of polynomial equations with some of the network parameters as variables and its coefficients are not trivial because $d_1 \neq d_2$ and because of Assumption 5. Therefore, it is satisfied only on a set of Lebesgue measure zero [Okamoto, 1973].

**Case 2.** Now assume $Ch_D \cap C = C$, then the factorization of Equation 13 simplifies to:

$$\prod_{C_i \in Ch_D} P(C_i \mid \overline{pa}_{C_i}, D = d_1; \sigma) \tag{18}$$

Therefore we can rewrite 12 as:

$$\sum_z U(d_1, x) \prod_{C_i \in Ch_D} P(C_i | \overline{pa}_{C_i}, D = d_1; \sigma) - U(d_2, x) \prod_{C_i \in Ch_D} P(C_i | \overline{pa}_{C_i}, D = d_2; \sigma) = 0 \tag{19}$$

That again is a polynomial equation in some of the network parameters and therefore the set of solutions has Lebesgue measure zero. Again, since the set of parameters that satisfies Equation 12 has measure zero along at least one dimension, the whole set has measure zero.

This implies that for almost all $P, U$ and any given local intervention $\sigma$ the optimal decision is unique.

$\square$

In the following lemma, we consider masking as a special case of a local intervention. A local intervention deterministically transforms the states of a random variable $X$ via a predefined function. Specifically, we can apply a function that maps all states of $X \in Pa_D$ to a masked state, rendering $X$ unobservable to the agent under this intervention. We denote by $Pa'_D \subseteq Pa_D$ the subset of variables that remain unmasked. Consequently, the set $Pa_D \setminus Pa'_D$ represents the variables that are masked and thus hidden from the agent when making the decision $D$.

**Lemma 2.** *Given a CID $M = (G, P)$, under Assumptions 1~9, given an optimal policy oracle $\Pi_\Sigma^*$ where $\Sigma$ includes all mixtures of local interventions on $C$ including masking inputs $Pa'_D \subseteq Pa_D$, then for any given $Pa'_D = pa'_D$ such that $Pa'_D \cap Pa_U = \emptyset$, we can identify:*

$$\sum_z P(C = c | do(D = d); \sigma) U(d, x) - P(C = c | do(D = d'); \sigma) U(d', x) \tag{20}$$

*for some $d, d' \in dom(D)$ where $d \neq d'$ and $\mathbf{Z} = C \setminus Pa'_D$.*

*Proof.* By Lemma 1, for almost all $P, U$ there exist only one optimal decision $d_1 = \arg\max_d \mathbb{E}[u | do(D = d), pa'_D; \sigma]$ following the shift $\sigma$. The decision $d_1$ can be identified using the optimal policy oracle $\Pi_\Sigma^*(\sigma)$.

By Assumption 9 we know that for every decision $d$ in the context $Pa'_D \subseteq Pa_D$, there exists at least one instance $c = (c_1, \ldots, c_N)$ of $C$ where $d \neq \arg\max_{d'} U(d', x)$. Note that we can set the values for $X$ as $Pa'_D \cap Pa_U = \emptyset$. So let $x'$ be the instantiation of $Pa_U \setminus \{D\}$ that satisfies $d_1 \neq \arg\max_d U(d, x')$, and $\sigma'$ be a hard intervention that sets $X$ to $x'$, then there exist $d_2 = \arg\max_d U(d, x')$, with $d_2 \neq d_1$. Note that under a hard intervention like $\sigma'$ we have $\mathbb{E}[u | do(D = d), pa'_D; \sigma] = U(d, x')$ where $x'$ are the values that the variables $Pa_U \setminus \{D\}$ take after the intervention. We can pick $\sigma'$ such that it sets $Pa'_D$ to be the same as in observation.

For $q \in [0, 1]$ consider the joint distribution over $C$ under the parametrized family of mixed local interventions $\tilde{\sigma}(q) = q\sigma + (1 - q)\sigma'$:

$$P(C = c | do(D = d); \tilde{\sigma}(q)) = qP(C = c | do(D = d); \sigma) + (1 - q)P(C = c | do(D = d); \sigma')$$

$$\tag{21}$$

Table 1: A partition of the CID's chance nodes $C$ that assigns them to cases in Theorem 2. Cells marked with (1) correspond to nodes that can be identified by case 1 of Theorem 1 in Richens and Everitt [2024]. Cells with (2) correspond to case 2. Cells with $\emptyset$ indicate that either no nodes fall within that intersection or that those nodes can be pruned since we are not able to learn the structure or the parameters for those. The cell marked with (3) corresponds to nodes that exist only in the mediated case and were therefore not considered by the previous paper.

|  | $(Anc_D \cup Desc_D)^C$ | $Anc_D$ | $Desc_D$ |
|---|---|---|---|
| $Anc_U(G_{\underline{D}})$ | (1) | (1) | (3) |
| $Anc_U(G_{\underline{D}})^C$ | $\emptyset$ | (2) | $\emptyset$ |

From Assumption 9 it follows that $Z := C \setminus Pa_D \neq \emptyset$. We can write the expected utility as:

$$\mathbb{E}[U|pa_d, do(D = d); \tilde{\sigma}(q))] = \sum_z P(Z = z|pa_D, do(D = d); \tilde{\sigma}(q))U(d, x) \qquad (22)$$

$$= \sum_z \frac{P(C = c|do(D = d); \tilde{\sigma}(q))}{P(Pa_D = pa_D|do(D = d); \tilde{\sigma}(q))}U(d, x) \qquad (23)$$

$$= \frac{1}{P(Pa_D = pa_D; \tilde{\sigma}(q))} \sum_z qP(C = c|do(D = d); \sigma)U(d, x) + $$
$$+ (1 - q)P(C = c|do(D = d); \sigma')U(d, x') \qquad (24)$$

Where in Equation 24 $P(Pa_D = pa_D|do(D = d); \tilde{\sigma}(q)) = P(Pa_D = pa_D; \tilde{\sigma}(q))$ according to Rule 3 of do-calculus since $D \perp\!\!\!\perp Pa_D$ in $G_{\overline{D}}$ [Pearl, 2009]. Note that $d_1$ is the optimal decision for $q = 1$, but that is not the case for $q = 0$. Therefore there exists $q_{crit}$ such that for all $q < q_{crit}$ $d_2 := \Pi_\Sigma^*(\tilde{\sigma}(q))$ is a decision in the set $\{d|d = \arg\max_d U(d, x')\}$, and for $q \geq q_{crit}$ the optimal decision is not in this set. Let $d_3 \notin \{d|d = \arg\max_d U(d, x')\}$. Consider the following equation:

$$\mathbb{E}[U|pa_D, do(D = d_2); \tilde{\sigma}(q_{crit})] - E[U|pa_D, do(D = d_3); \tilde{\sigma}(q_{crit})] = 0 \qquad (25)$$

$$\iff \quad q_{crit}\left[\sum_z P(C = c|do(D = d_2); \sigma)U(d_2, x) - P(C = c|do(D = d_3); \sigma)U(d_3, x)\right] + $$
$$+ (1 - q_{crit})[U(d_2, x') - U(d_3, x')] = 0 \qquad (26)$$

$$\iff$$

$$q_{crit} = \left(1 - \frac{\sum_z P(C = c|do(D = d_2); \sigma)U(d_2, x) - P(C = c|do(D = d_3); \sigma)U(d_3, x)}{U(d_2, x') - U(d_3, x')}\right)^{-1} \qquad (27)$$

Therefore, since the functional relationship between $U$ and its parents is known, if we find $q_{crit}$ we can identify:

$$\sum_z P(C = c|do(D = d_2); \sigma)U(d_2, x) - P(C = c|do(D = d_3); \sigma)U(d_2, x) \qquad (28)$$

$\square$

Let $G_{\underline{D}}$ be $G$ without the edges leaving $D$, the mediated case allows for $Desc_D \cap Anc_U(G_{\underline{D}}) \neq \emptyset$. Consider the partition of $C$ proposed in Table 1, the proof of Theorem 1 in Richens and Everitt [2024] can still be used with minor changes in the mediated case for some of the nodes, but a new case needs to be introduced for $Anc_U(G_{\underline{D}}) \cap Desc_D$.

**Theorem 2.** *Let $M$ be a CID satisfying Assumptions 1~9. Suppose for each $C_i$, we know a set $\widehat{Pa}_{C_i} \subseteq Pa_{C_i}$ and a set of nodes $V_{kwn} \subseteq V$ where $C_i \in V_{kwn} \iff \widehat{Pa}_{C_i} = Pa_{C_i}$. Let $\{\pi^*_\sigma(d|pa_D)\}_{\sigma \in \Sigma}$ denote the set of optimal policies available where $\pi^*_\sigma(d|pa_D)$ is an optimal policy under the intervention $\sigma$ and $\Sigma$ is the set of all mixtures of local interventions.*
*Then, for almost all CIDs, the graph $G$ and the joint distribution $P$ over all the ancestors of the utility node $Anc_U$ can be identified.*

*Proof.* Let $G_{\underline{D}}$ be $G$ without the edges leaving $D$. Following the CID's chance node partition described in Table 1, we consider three cases:

- [Case 3, $Anc_U(G_{\underline{D}}) \cap Desc_D$]. For the third case, we provide a constructive proof for nodes in $Anc_U(G_{\underline{D}}) \cap \overline{Desc}_D$. We establish this proof by strong induction. Consider a directed path $C_k \to \cdots \to C_1$ where $C_1 \in Pa_U, \forall i. C_i \in Desc_D \setminus Ch_D$. Assume we know $Pa_{C_i}$ and $P(C_k|Pa_{C_i})$ for all $i = 1, \ldots, k-1$, we want to learn $Pa_{C_k}$ and $P(C_k|Pa_{C_k})$. For each of the nodes $Y_i$ in $Y := C \setminus \{C_1, \ldots, C_k\}$ we define the following hard interventions $\sigma_{C_k}(Y \setminus Y_i = y, Y_i = \kappa) := do(Y_1 = y_1, \ldots, Y_i = \kappa, \ldots, Y_{|Y|} = y_n, C_k = f(c_k))$ where $y$ is an instantiation for $Y \setminus \{Y_i\}$ and $\kappa$ one for $Y_i$. Here $f(C_k)$ is the following local intervention on $C_k$:

$$f(C_k) = \begin{cases} c'_k, & C_k = c'_k \\ c''_k, & \text{otherwise} \end{cases} \tag{29}$$

We also mask all inputs to the policy: $Pa'_D = \emptyset$. Assume $C_k \notin Ch_D$, by Lemma 2 we can identify the following query:

$$\sum_c P(C = c|do(D = d); \sigma_{C_k}(Y \setminus Y_i = y, Y_i = \kappa))U(d, x) - $$
$$-P(C = c|do(D = d'); \sigma_{C_k}(Y \setminus Y_i = y, Y_i = \kappa))U(d', x) = \tag{30}$$

$$= \sum_{c_k} \cdots \sum_{c_1} (\prod_{j=1}^k P(C_j = c_j|pa_{C_j}, do(D = d); \sigma_{C_k}(Y \setminus Y_i = y, Y_i = \kappa))U(d, x) - $$
$$- \prod_{j=1}^k P(C_j = c_j|pa_{C_j}, do(D = d'); \sigma_{C_k}(Y \setminus Y_i = y, Y_i = \kappa))U(d', x)) \tag{31}$$

According to Rule 3 of do-calculus [Pearl, 2009], since $D \perp\!\!\!\perp C_1, \ldots, C_k|Y$ in $G_{\overline{Y}}$ the expression in Equation 31 is equal to:

$$= \sum_{c_k} \cdots \sum_{c_1} \prod_{j=1}^k P(C_j = c_j|pa_{C_j}; \sigma_{C_k}(Y \setminus Y_i = y, Y_i = \kappa))[U(d, x) - U(d', x)] \tag{32}$$

$$= \sum_{c_k} P(C_k = c_k|pa_{C_k}; \sigma_{C_k}(Y \setminus Y_i = y, Y_i = \kappa))\beta(c_k) \tag{33}$$

$$= \sum_{c_k} P(C_k = c_k|pa_{C_k})\beta(c_k) \tag{34}$$

where:

$$\beta(c_k) := \sum_{c_{k-1}} \cdots \sum_{c_1} \prod_{j=1}^{k-1} P(C_j = c_j|pa_{C_j}; \sigma_{C_k}(Y \setminus Y_i = y, Y_i = \kappa))[U(d, x) - U(d', x)] \tag{35}$$

This result is analogous to the one for Case 1. In Equation 34, following the definition of the intervention $\sigma_{C_k}(Y \setminus Y_i = y, Y_i = \kappa)$, we have:

$$P(C_k = c'_k|pa_{C_k}; \sigma_{C_k}(Y \setminus Y_i = y, Y_i = \kappa)) = P(C_k = c_k|pa_{C_k}) \tag{36}$$

and

$$P(C_k = c''_k|pa_{C_k}; \sigma_{C_k}(Y \setminus Y_i = y, Y_i = \kappa)) = $$
$$= 1 - P(C_k = c'_k|pa_{C_k}; \sigma_{C_k}(Y \setminus Y_i = y, Y_i = \kappa)) = \tag{37}$$
$$= P(C_k = c_k|pa_{C_k})$$

Therefore there is only one parameter to be identified. If $C_k \notin V_{kwn}$, we can repeat this procedure with a different leave-one-out intervention for each potential parent $Y_i$ of $C_k$ and different $c_k$. If for some configuration of $\kappa_1, \kappa_2$ and $c_k$ we have

$$P(C_k = c_k' | pa_{C_k}; \sigma_{C_k}(Y \setminus Y_i = y, Y_i = \kappa_1))$$
$$\neq$$
$$P(C_k = c_k' | pa_{C_k}; \sigma_{C_k}(Y \setminus Y_i = y, Y_i = \kappa_2))$$

then $Y_i \in Pa_{C_k}$. We can exclude from this search all nodes in $\widehat{Pa}_{C_k}$, since we already know they are parents of $C_k$. If $C_k \in V_{kwn}$ we can skip this step since we already know $Pa_{C_k}$. Then, for each instantiations of the variables in $Pa_{C_k}$ and each $c_k \in dom(C_k)$ we repeat the procedure and recover all the parameters for $C_k$.

And now we describe the necessary modification to the cases covered in Richens and Everitt [2024] (Case 1 and 2 in Table 1):

1. $Anc_U(G_{\underline{D}}) \cap \left[(Anc_D \cup Desc_D)^C \sqcup Anc_D\right]$. The identification problem for nodes in this set is described in Theorem 1 Case 1 of Richens and Everitt [2024]. The proof is based on strong induction on $k$ for directed paths $C_k \to \cdots \to C_1$ where $C_1 \in Pa_U$, where for all $i = 1, \ldots, k$ we have $C_k \neq D$. The procedure to incorporate prior knowledge is the same as the one specified in the proof for Case 3.

2. $Anc_U(G_{\underline{D}})^C \cap Anc_D$. This case corresponds to Theorem 1 Case 2 of Richens and Everitt [2024]). The original proof considered strong induction on $k$ for directed paths $C_k \to \cdots \to C_1$ where $C_1 \in Pa_D$. Again, the procedure to incorporate prior knowledge is the same as the one specified in the proof for Case 3.

$\square$

## A.3 Algorithmic Complexity of LearnCID

First consider that the complexity of LearnCID (Algorithm 1) depends on the complexity of querying the policy oracle, let us call this complexity $K$. As a worst-case scenario there is no prior knowledge about the graph, so $V_{kwn}$ is empty. Let $n$ be the number of variables, and $b := \max_{X \in C} |dom(X)|$ be the maximum number of observable values of any chance variable. The most computationally expensive steps correspond to computing $q_{crit}$ with $ALG_{q_{crit}}$ (line 10) and the CPTs' entries for all the chance nodes. Each call to $ALG_{q_{crit}}$ costs $O(N(K + |dom(D)|))$, filling one CPT's entry costs $O(nb^n)$. Overall the algorithm's time complexity is $O(n^2 b^n N(K + |dom(D)|) + n^3 b^{2n})$.

## A.4 Comparison to Richens and Everitt [2024]

Richens and Everitt introduced an innovative perspective on the connection between robustness to distribution shifts and causal understanding of the environment [Richens and Everitt, 2024]. While the existence of a link between causal reasoning and generalization across domains was well established [Pearl and Bareinboim, 2011, Schölkopf et al., 2021], it was unclear whether causal understanding was necessary for domain adaptation. They were the first to show that agents must learn a causal model of their environment to achieve robustness to distribution shifts. This work builds on and extends their contributions in several ways. First, while their results are limited to unmediated tasks, this work shows that causal information is learned by robust agents even in mediated settings. Second, this paper includes a general algorithm for eliciting CIDs from optimal policy oracles, whereas Richens and Everitt [2024] presents an example limited to a two-variable CID. Third, the main theorem and algorithm of this paper explicitly allow for incorporating prior knowledge about the CID's causal structure, which is a dimension not addressed in their work. Fourth, the analysis is extended to multi-agent systems and techniques are discussed for applying the LearnCID algorithm in such settings. Finally, while Richens and Everitt [2024] focuses on single-shot tasks, this work also explores sequential decision-making by representing POMDPs as infinite CIDs, drawing on prior structural ideas from Everitt and Bareinboim [Everitt et al., 2019, Bareinboim et al., 2024], and shows how LearnCID can be applied to time-homogeneous POMDPs.

# B   Non-identifiability of $Ch_D$ and their CPTs

Now we prove that, under the same assumptions made for the previous results, $Ch_D$ cannot be uniquely determined in general. This motivates the LearnCID assumption requiring knowledge of the children of $D$, their CPTs, and their parents.

**Theorem 3.** *Let $M = (G = \{V, E\}, P)$ be a single decision/single utility CID, assume we know $D, U, C,$ and $Pa(U), Pa(D)$. Let $\Sigma$ be the set of all mixtures of local interventions, $\Pi^*_\Sigma$ be an optimal policy oracle for $D$. Then in general $Ch_D$ cannot be uniquely determined.*

*Proof.*  Consider the following example:

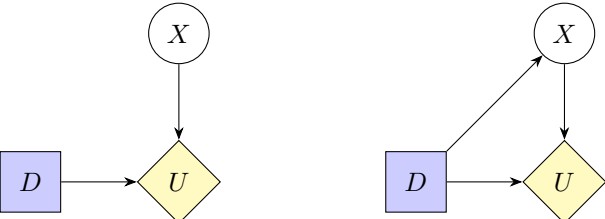

Figure 4: On the left, a CID where $X \notin Ch_D$. On the right, a CID with the same nodes, but where $X \in Ch_D$.

Let $X$ and $D$ be binary variables and consider the CID illustrated on the left side Figure 4. Let $p := P(X = 1) = 0.9$ and $P(X = 0) = 0.1$. Let $U(d, a) := 1$ if $d = a$ and 0 otherwise. In this case the optimal decision without interventions is $d^*$ is 1. There are four possible local interventions $\sigma_1 := do(X = 1)$, $\sigma_0 := do(X = 0)$, $\sigma_{id}$ corresponding to an identity map for $X$, $\sigma_s$ switching the values 0 and 1 of $X$. The set of all mixtures of local interventions $\Sigma$ includes all the interventions in the parametrized family of mixtures $\sigma(p_0, p_1, p_{id}, ps) := p_0\sigma_0 + p_1\sigma_1 + p_{id}\sigma_{id} + p_s\sigma_s$ with all the convex coefficient $p_i \in [0, 1]$ and $0 \le p_i \le 1$. Note that each of the four local interventions is a special case of this mixture, obtained by setting the corresponding coefficient to 1 and the others to 0.

| $D$ | $X$ | $P(X \mid D)$ |
|---|---|---|
| 0 | 0 | 0.2 |
| 0 | 1 | 0.8 |
| 1 | 0 | 0 |
| 1 | 1 | 1 |

Table 2: Conditional probabilities of $X$ given $D$ for the CID where $D \in Ch_D$.

Also consider the CID illustrated on the right side of Figure 4, along with its corresponding conditional probability table (CPT) in Table 2. The set of chance nodes is the same in both CIDs; therefore, the set of all possible mixtures of local interventions, denoted $\Sigma$, is also identical. For any given pair of interventions, the critical value $q_{crit}$ is the same in both models. For example, the policy oracle $\Pi^*_\Sigma$ returns $d^* = 0$ for $\sigma_0$, $d^* = 1$ for $\sigma_1$, and for $\sigma(q)$, it returns $d^* = 0$ when $q < 0.5$, $d^* = 1$ when $q > 0.5$, and may return any policy when $q = q_{crit} = 0.5$.

The same behavior holds for other mixtures, such as:

- $\sigma'(q) := q\sigma_1 + (1 - q)\sigma_s$, where $q_{crit} = \frac{4}{9}$,

- $\sigma''(q) := q\sigma_{id} + (1 - q)\sigma_s$, where again $q_{crit} = 0.5$.

In general, for all mixtures in $\Sigma$, both the critical threshold $q_{crit}$ and the optimal decision returned by the oracle are the same in both models. Therefore, it is not possible to distinguish between the two CIDs based on oracle responses alone.

Now, let $z := P(X = 1 \mid D = 1)$ and $q := P(X = 1 \mid D = 0)$ in the second CID. Observe that regardless of the specific values of $z$ and $q$, the model is equivalent to a model with $X \notin \mathrm{Ch}(D)$ and marginal distribution $P(X = 1) = \frac{z+q}{2}$. $\qquad \square$

We also show that unlike for other chance nodes, the CPT of the children chance variables of the decision node can not be fully estimated in the general case. It might be possible to estimate only those parts of the CPT that correspond to decisions that are optimal for some distribution shift $\sigma$, but not for the others.

**Corollary 1** (of Lemma 2). *Let $M = (G = \{V, E\}, P)$ be a single decision/single utility CID, assume we know $G$. Let $\Sigma$ be the set of all mixtures of local interventions, $\Pi_\Sigma^*$ be the optimal policy oracle. Then, using the identification result of Lemma 2, in general the CPTs for $C \cap Ch_D$ cannot be uniquely determined for a set of parameters with a strictly positive Lebesgue measure.*

*Proof.* Consider the CID described on the right side of Figure 4 with the CPT described in Table 3. Assume $dom(D) = \{0, 1, 2\}$ and that we don't know the CPT for $X$. The set of local interventions $\Sigma$ contains $\sigma_1 := do(X = 1)$, $\sigma_0 := do(X = 0)$, $\sigma_{id}$ corresponding to an identity map for $X$, and the parametrized family of mixtures $\sigma(p_0, p_1, p_{id}, ps) := p_0\sigma_0 + p_1\sigma_1 + p_{id}\sigma_{id} + p_s\sigma_s$ with all the convex coefficient $p_i \in [0, 1]$ and $0 \le p_i \le 1$. For all the feasible combinations of $p_0, p_1, p_{id}, p_s$ the deterministic policy $d = 2$ is dominated by $d = 0$ and $d = 1$.

| $D$ | $X$ | $P(X \mid D)$ |
|-----|-----|---------------|
| 0 | 0 | 0.5 |
| 0 | 1 | 0.5 |
| 1 | 0 | 0 |
| 1 | 1 | 1 |
| 2 | 0 | 0.4 |
| 2 | 1 | 0.6 |

Table 3: Conditional probabilities of $X$ given $D$, in this example $dom(D) = \{0, 1, 2\}$.

Therefore, for all $\sigma \in \Sigma$, the deterministic policy corresponding to $d = 2$ is never selected by the policy oracle $\Pi_\Sigma^*(\sigma)$. Since the identification result of Lemma 2 includes only probabilities $P(C \mid do(D = d'); \sigma)$ where $d'$ is an optimal solution for some $\sigma$ selected by the policy oracle, it follows that, in particular, we cannot identify $P(X \mid do(D = 3)) = P(X \mid D = 3)$, or, more generally, the portion of the CPT for chance variables that are children of $D$ corresponding to decisions that are never optimal under any intervention $\sigma$. $\qquad \square$

## C  Domain dependence in mediated tasks

Previous results [Richens and Everitt, 2024] show that for unmediated decision tasks domain dependence implies Assumption 9. Here we show that this implication does not hold in the mediated case. Moreover, we prove that Assumption 9 implies domain dependence in the mediated case and therefore that Assumption 9 is equivalent to domain dependence in the unmediated case, which is a subcase of the mediated one. We report the definition of Domain dependence.

**Definition 2** (Domain dependence [Richens and Everitt, 2024]). A CID $M$ is said to satisfy domain dependence if there exist $P(C = c)$, $P'(C = c)$, both compatible with the CID $M$ such that $\pi^* = \arg\max_\pi \mathbb{E}_P^\pi[U] \implies \pi^* \ne \arg\max_\pi \mathbb{E}_{P'}^\pi[U]$.

| D | X | U(X\|D) |
|---|---|---|
| 0 | 0 | 0 |
| 0 | 1 | 1 |
| 1 | 0 | 0 |
| 1 | 1 | 2 |

| D | X | P(X\|D) | P'(X\|D) |
|---|---|---|---|
| 0 | 0 | 0.5 | 0 |
| 0 | 1 | 0.5 | 1 |
| 1 | 0 | 0 | 1 |
| 1 | 1 | 1 | 0 |

Figure 5: An example CID to show that in the mediated case domain dependence does not imply Assumption 9. Starting from the left, a specification of the utility function associated with node $U$, the example's CID, and the two CPTs for $X$ before the distribution shift ($P$) and after the distribution shift ($P'$).

Consider the example described in Figure 5. Let $U(d,x) := 2$ if $d = x = 1$, $U(d,x) := 1$ if $d = 0$ and $x = 1$, and 0 otherwise. For the distribution $P$ the only optimal policy corresponds to always choosing $D = 1$ because $X$ will always correspond to 1 and therefore the expected utility is 2. But this policy is no longer optimal under $P'$ because $X$ will always be observed as 0 and the expected utility is 0 while for example a policy that always chooses $D = 0$ corresponds to an expected utility of 1. Therefore domain dependence holds, but at the same time, Assumption 9 does not hold because $d^* = 1 \in \arg\max_d U(d,x)$ for all $x \in dom(X)$. Therefore, domain dependence $\not\Longrightarrow$ Assumption 9 in the general mediated case.

Now we prove that Assumption 9 implies domain dependence in the mediated case, and consequently is equivalent to domain dependence in the unmediated case.

**Theorem 1.** *Let $M = (G, P)$ be a CID where $Desc_D \cap Anc_U \neq \emptyset$ (mediated task).*

$$\text{Assumption 9} \implies \text{Domain dependence.}$$

*Proof.* Assume $\forall\ P'(C = c)$ compatible with $M$ we have $\pi^* \in \arg\max_\pi \mathbb{E}[U] = \arg\max_\pi \mathbb{E}^\pi_{P'}[U|do(D = \pi(d|pa_D)), pa_D]$. Let $d \in dom(D)$ be a decision s.t. $\pi^*(d|pa_D) > 0$. For Assumption 9 there exist a non-empty set $X_d := \{x | d \notin \arg\max_{d'} U(d', x)\}$. Let $d^*, x^* \in \arg\max_{d', x \in X_d} U(d', x)$. We can write $x^*$ as $(x_1^*, \ldots, x_n^*)$ where $\{x_i^*\}_{i=1}^n$ are instantiations of the random variables $\{X_1, \ldots, X_n\} = Pa_U \setminus \{D\}$ and $n := |Pa_U \setminus \{D\}|$. Now, we want to define an alternative distribution $P'$ compatible with $M$ by updating the CPTs of the variables corresponding to the parents of $U$. For each $X_j \in Pa_u \setminus \{D\}$ let $pa_j^1, \ldots, pa_j^{|Pa_{X_j}|}$ be instantiations of parents of $X_j$. Let $x_j^i$ be an observable value for the variable $X_j$. We set $P(x_j^i|pa_j^1, \ldots, pa_j^{|Pa_{X_j}|}) = \epsilon_{pa_j^1, \ldots, pa_j^{|Pa_{X_j}|}, x_j^i}$ if $x_j^i \neq x_j^*$ and $P(x_j^i|pa_j^1, \ldots, pa_j^{|Pa_{X_j}|}) = 1 - \sum_{l \neq i} \epsilon_{pa_j^1, \ldots, pa_j^{|Pa_{X_j}|}, x_j^l}$ if $x_j^i = x_j^*$. We repeat this procedure for all combinations of $x_j^i$ and $pa_j^1, \ldots, pa_j^{|Pa_{X_j}|}$. We call the set of these epsilon parameters $\Sigma$. To preserve faithfulness we require the epsilon parameters to be pair-wise distinct. Observe that if we also require all epsilon $\epsilon \in \Sigma$ to be $0 < \epsilon \ll 1$ and $0 < \sum_{\epsilon \in \Sigma} \epsilon \leq 1$ then we obtain valid CPT parameters for the CID. We repeat the CPT update for all variables $\{X_1, \ldots, X_n\}$.

Observe that assuming $\pi^* \in \arg\max \mathbb{E}^\pi_{P'}[U]$ for all $P'$ compatible with $M$ implies that $\forall\ \pi'$ and $P'$ compatible with $M$ we have:

$$\mathbb{E}^{\pi^*}_{P'}[U] - \mathbb{E}^{\pi'}_{P'}[U] \geq 0 \tag{38}$$

$$\iff \mathbb{E}_{P'}[U \mid do(D = \pi^*(d \mid pa_D) =, pa_D] - \mathbb{E}_{P'}[U \mid do(D = \pi'(d \mid pa_D), pa_D] \geq 0 \tag{39}$$

Let $\pi'$ be a deterministic policy where $d^*$ is always selected and $X := Pa_U \setminus \{D\}$ with $x$ instantiation of $X$.

$$\iff \mathbb{E}_{P'}[U \mid do(D = \pi^*(d \mid pa_D) =, pa_D] - \mathbb{E}_{P'}[U \mid do(D = d^*), pa_D] \geq 0 \tag{40}$$

$$\iff \sum_{d'} \pi^*(d' \mid pa_D) \sum_{C_i \in \mathbf{C}} \prod_j P(C_i = c_j \mid pa_{C_i}) U(d', x) - \\ - \sum_{C_i \in \mathbf{C}} \prod_j P(C_i = c_j \mid pa_{C_i}) U(d^*, x) \geq 0 \tag{41}$$

Now we compute the limit for all $\Sigma \ni \epsilon \to 0$:

$$\lim_{\Sigma \ni \epsilon \to 0} \mathbb{E}_{P'}^{\pi^*}[U] - \mathbb{E}_{P'}^{\pi'}[U] = \tag{42}$$

$$= \sum_{d'} \pi^*(d' \mid pa_D)U(d', x^*) - U(d^*, x) \tag{43}$$

$$= \pi^*(d \mid pa_D)U(d, x^*) + \sum_{d' \neq d} \pi^*(d' \mid pa_D)U(d', x) - U(d^*, x^*) \leq \tag{44}$$

$$\leq \pi^*(d \mid pa_D)U(d, x^*) + U(d^*, x^*) \left[ \sum_{d' \neq d} \pi^*(d' \mid pa_D) - 1 \right] \tag{45}$$

where in the last passage we used the fact that $d^* \in \arg\max_{d'} U(d', x^*)$.

$$= - \left[ \sum_{d' \neq d} \pi^*(d' \mid pa_D) - 1 \right] U(d, x^*) + U(d^*, x^*) \left[ \sum_{d' \neq d} \pi^*(d' \mid pa_D) - 1 \right] \tag{46}$$

$$= \pi^*(d \mid pa_D) \left( U(d^*, x^*) - U(d, x^*) \right) < 0 \tag{47}$$

The last expression is strictly negative because we assumed $\pi^*(d \mid pa_D) > 0$ and $x^* \in X_d$, therefore $d \notin \arg\max_{d'} U(d', x^*)$ while $d^* \in \arg\max_{d'} U(d', x^*)$. Since $\mathbb{E}_{P'}^{\pi^*}[U] - \mathbb{E}_{P'}^{\pi'}[U]$ is a polynomial in the parameters $\Sigma$, we can apply the theorem of permanence of sign and therefore $\exists \epsilon^* \in \Sigma$ s.t. the inequality 38 is false. Therefore $\exists P'$ compatible with $M$ s.t. $\pi^* \notin \arg\max_\pi \mathbb{E}_{P'}^\pi \not\downarrow$. It follows that domain dependence holds. $\square$

From Theorem 1 it directly follows that for unmediated decision tasks, which are a subcase of the family of mediated decision tasks, domain dependence is equivalent to Assumption 9. This provides us with a very straightforward way to verify domain dependence in these tasks.

**Corollary 2.** *Let $M = (G, P)$ be an unmediated CID. Assumption 9 is equivalent to Domain dependence.*

*Proof.* In the unmediated case the implication Domain dependence $\implies$ Assumption 9 is proven in Richens and Everitt [2024]. Since the unmediated case is a subcase of the mediated case, from Theorem 1 it directly follows that Assumption 9 $\implies$ Domain dependence. Therefore the two statements are equivalent. $\square$

## D  Example of mixture of local interventions

As an example of how a mixture of local interventions can represent a distribution shift, consider the following (illustrated in Figure 6): we have a random variable $X$ representing the outcome of a biased coin flip that always lands on heads, where $X \in \{H, T\}$ corresponds to heads and tails, respectively. Let $\sigma_i := do(X = i)$ with $i \in \{H, T\}$, then define $\sigma^* = \sum_i p_i \sigma_i$. By changing the coefficients $p_i$, we can map the distribution of the fair coin to any distribution on the observable values set $dom(X) = \{H, T\}$. For example, by setting $p_H = \frac{2}{3}$ and $p_T = \frac{1}{3}$, we can map the original distribution to a new one where heads is observed $\frac{2}{3}$ of the time and tails is observed $\frac{1}{3}$ of the time. Note that in this example, each local intervention was a hard intervention because, regardless of the value of the coin, each intervention mapped it to a specific value. In general, this is not required, as a local intervention can be any deterministic map from the set of observable values to itself.

While hard interventions break the dependence of a variable on its parents, local interventions may preserve this dependence. For example, consider an unbiased coin flip $Y$ and a die throw $X$. If the coin lands on heads, we throw a six-sided die, if it lands on tails, we throw a twelve-sided die. In this scenario, $Y$ and $X$ are clearly dependent and in the corresponding causal graph $Y \in Pa_X$. Now consider the local intervention $\sigma_m := do(X = X \mod 12 + 1)$. This intervention increases the die result by one or sets it to 1 if the result was 12. This local intervention does not make $X$ independent of $Y$ because, for instance:

$$P(X = 4 | Y = T; \sigma_m) = P(X = 3 | Y = T) = \frac{1}{12} \tag{48}$$

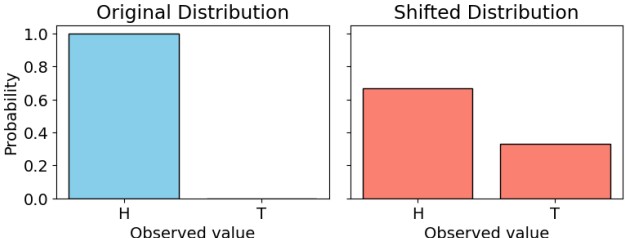

Figure 6: Histograms illustrating the distribution shift induced by a mixture of local interventions. The left histogram shows a biased coin distribution $P(X)$, where the coin always lands on heads. The right histogram represents a shifted distribution $P(X \mid \sigma^*)$ obtained by applying the mixture $\sigma^* = p_H \sigma_H + p_T \sigma_T$ for $\sigma_i := do(X = i)$ with $p_H = \frac{2}{3}$ and $p_T = \frac{1}{3}$.

However,

$$\frac{1}{12} \neq \frac{1}{8} = P(X = 3) = P(X = 4|\sigma_m) \tag{49}$$

# E  LearnCID examples

In this section, we provide two examples to illustrate the application of LearnCID (Algorithm 1) in both single and multi-agent environments.

## E.1  Example 1 - Single-agent environment

Now we will go over a complete, step-by-step example of a LearnCID application to learn the simple CID illustrated in Figure 7. For this example, we assume $V_{kwn}$ is empty.

First, let us go over the LearnCID Assumptions 1-9:

1. By observing the (unknown) CPT of $A$ we can see that for this example faithfulness holds, there are no latent confounders so causal sufficiency also holds.

2. The set of decision nodes is $\{D\}$, the utility node associated with the utility function $U(d, a)$ is $U$, and the set of chance nodes is $C = \{A, B\}$.

3. The CID indeed contains one decision node $D$ and one utility node $U$.

4. The Markov blanket of $D$ is $\{B\}$, and we know that $D$ is a parent of $B$, i.e., the graph contains the edge $D \to B$.

5. Since the decision is an input of the utility function $U(d, a)$ we know $D$ is a parent of $U$.

6. Both chance nodes $A$ and $B$ are ancestors of $U$.

7. The utility function $U(d, a) := 1$ if $d = a$ and 0 otherwise, is fully specified.

8. We have an optimal policy oracle $\Pi^*_\Sigma$ where $\Sigma$ is the set of all mixtures of local interventions.

9. We can verify that for $A = 1$, $\arg\max_d U(d, A = 1) = 1$ but $\arg\max_d U(d, A = 0) = 0$.

For ease of comprehension we summarize the application of Algorithm 1 to the example CID in Figure 7, in the following steps:

S1  Visit $A$, the only unvisited chance node with a known path to $U$.

S2  Define local interventions on $B$ as in Equation 4.

S3  Estimate $q_{crit}$ using ALG$q_{crit}$ (Algorithm 1 in Richens and Everitt [2024]).

S4  Compute $P(A = a_i|do(B = b_i))$.

S5  Repeat steps 2 to 4 for all configurations of $a_i$ and $b_i$.

S6  Deduce the set of parents of $A$ and its CPT.

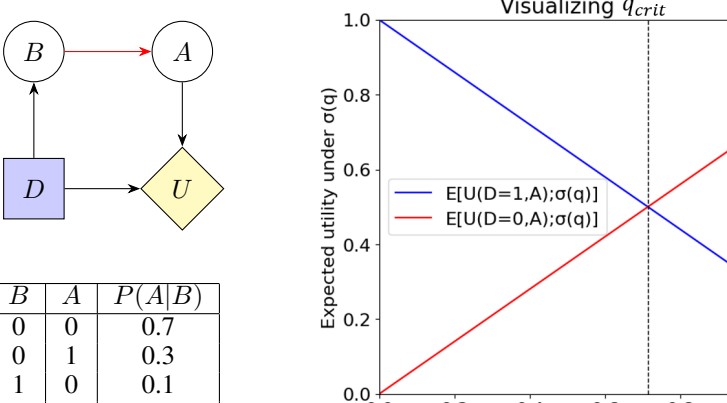

Figure 7: An example of a single-decision/single-utility CID. On the bottom-left, the CPT for variable $A$. The edge marked in red is unknown. On the right, following Example 1, let us examine two interventions: $\sigma_0 := do(B = 0)$ with optimal decision $d_1$, and a hard intervention $\sigma'$ with optimal decision $d_2$ where $d_1 \neq d_2$. We define a mixture of local interventions as $\sigma(q) = q\sigma_0 + (1 - q)\sigma'$. The plot displays the expected utility of both decisions as $q$ varies. The value for $q$ where both decisions become simultaneously optimal is called $q_{crit}$.

Following the aforementioned steps:

**Step 1**: In this example $Pa_D$ is empty and $Pa_U = \{A\}$. LearnCID visits and finds the set of parents and CPT of all chance nodes that have a known path to $U$, since at the beginning the only chance node with a known path to $U$ is $A$, we start by visiting node $A$ (line 3).

Observe that since $A$ is the only chance node that is not children of $D$ the process will stop after $A$. On line 4, the variable "Path" is initialized with a set of all chance nodes in one path from $A$, the chance node we are currently visiting, and $U$. Since $A \in Pa_U$, "Path" is empty.

On line 5, we initialize the set $C_A$ by including all the chance nodes that are not $A$ and not in the path from $A$ to $U$ we selected ("Path"). Since "Path" is empty, the only chance node different from $A$ is $B$ and therefore $C_A = \{B\}$.

**Step 2**: Now, from $C_A = \{B\}$ we pick a node that could potentially be a parent of $A$ (lines 6 and 8), in this case the only option is $B$. Since $C_A \setminus \{B\}$ is empty, then the only possible instantiation for $C_A \setminus \{B\}$ is $\emptyset$. Then on line 9, following Equation 4, we initialize a set of interventions $\sigma_B(\emptyset)$ as:

$$\sigma_B(\emptyset) = \{\sigma_0, \sigma_1\} \qquad \text{where } \sigma_0 := do(B = 0), \text{and } \sigma_1 := do(B = 1) \tag{50}$$

Also, we pick an observable value for $A$ (line 7), for example 0, and aim to recover $P(A = 0; \sigma)$ for all $\sigma \in \sigma_B$. Note that since $A$ is binary, the function $f$ in Equation 4 is the identity function and can therefore be ignored.

**Step 3**: For each $\sigma \in \sigma_B$, in ALG$q_{crit}$ we use the oracle to find the optimal decision under $\sigma$ (line 11). For example, for $\sigma_0 = do(B = 0)$, $\Pi^*_\Sigma(\sigma_0)$ returns the optimal decision $d_0 := 0$. This is evident from the (unknown) CPT because for $B = 0$ the probability that $A = 0$ is higher than the one for $A = 1$, since the utility is maximized when $A$ and $D$ take the same value, the oracle returns the optimal decision $D = 0$ which will be equal to $A$ more often than $D = 1$.

ALG$q_{crit}$ then estimate $q_{crit}$. It works by finding an intervention $\sigma'$ such that the optimal decision is no longer $D = 0$ as it was under $\sigma_0$. Since in this example $D$ is binary, we already know the new optimal decision must be $D = 1$, in general, for this we can use the optimal policy oracle. We can find $\sigma'$ by hard intervening on the parents of the utility node $U$, which is $A$ in this case, such that $d_0$ is no longer optimal. Specifically, we can define $\sigma'$ as a hard intervention that sets $A$ to 1, therefore the new optimal decision is $d_1 := 1$. Note that this is always possible thanks to Assumption 9. Then, we define the mixture of local interventions $\sigma(q) := q\sigma_0 + (1 - q)\sigma'$. The right side of Figure 7 shows how the expected utility of both decisions varies with $q$. We can sample $q$ uniformly in the

interval $[0, 1]$ $N$ times and each time query the optimal policy oracle. Each time the oracle returns an optimal decision for the intervention $\sigma'$ we increment a counter $\theta$. Then $\frac{\theta}{N}$ is an unbiased estimate for $q_{crit}$. For this example, $q_{crit} = \frac{5}{7}$.

**Step 4**: Now we can compute $P(A = 0; \sigma_0)$. Consider the mixture of interventions $\sigma(q)$ from Step 3. We can write:

$$\mathbb{E}[U \mid do(D = 0); \sigma(q)] = \sum_{a,b} P(A = a, B = b \mid do(D = 0); \sigma(q))$$

$$= \sum_{a,b} q P(A = a, B = b \mid do(D = 0); \sigma_0) + (1 - q) P(A = a, B = b \mid do(D = 0); \sigma')$$

When $q = q_{crit}$ both decisions $D = 1$ and $D = 0$ are optimal:

$$E[U \mid do(D = 1); \sigma(q_{crit})] = E[U \mid do(D = 0); \sigma(q_{crit})]$$
$$\iff E[U \mid do(D = 1); \sigma(q_{crit})] - E[U \mid do(D = 0); \sigma(q_{crit})] = 0$$

$$\iff q_{crit} \left[ \sum_{a,b} P(A = a, B = b \mid do(D = 1); \sigma_0) U(1, a) - P(A = a, B = b \mid do(D = 0); \sigma_0) U(0, a) \right]$$
$$+ (1 - q_{crit}) \left[ U(d = 1, a = 1) - U(d = 0, a = 1) \right] = 0 \tag{51}$$

Recall $\sigma_o = do(B = 0)$. Since we know that $A$ is not a child of $D$ and we are hard intervening on all chance nodes non-descendants of $A$, by rule 3 of do-calculus [Pearl, 2009] $P(A = a \mid do(D = d, B = b)) = P(A = a \mid do(B = b))$, and we can write:

$$q_{crit} \left[ \sum_a P(A = a \mid do(D = 1, B = 0)) U(1, a) - P(A = a \mid do(D = 0, B = 0)) U(0, a) \right] +$$
$$+ (1 - q_{crit})[U(1, 1) - U(0, 1)] = 0$$
$$\iff q_{crit} \left[ \sum_a P(A = a \mid do(B = 0)) U(1, a) - P(A = a \mid do(B = 0)) U(0, a) \right] +$$
$$+ (1 - q_{crit})[U(1, 1) - U(0, 1)] = 0$$
$$\iff q_{crit} \left[ \sum_a P(A = a \mid do(B = 0))[U(1, a) - U(0, a)] \right] + (1 - q_{crit})[U(1, 1) - U(0, 1)] = 0$$

Now we can define the following expression, which corresponds to Equation 5:

$$\beta(a) := U(1, a) - U(0, a) \tag{52}$$

And rewrite Equation 51 as:

$$q_{crit} \left[ \sum_a P(A = a \mid do(B = 0)) \beta(a) \right] + (1 - q_{crit})[U(1, 1) - U(0, 1)] = 0$$
$$\iff q_{crit} \left[ P(A = 0 \mid do(B = 0)) \beta(0) + P(A = 1 \mid do(B = 0)) \beta(1) \right] +$$
$$+ (1 - q_{crit})[U(1, 1) - U(0, 1)] = 0$$

$$\iff q_{crit} \left[ P(A = 0 \mid do(B = 0)) \beta(0) + [1 - P(A = 0 \mid do(B = 0))] \beta(1) \right] +$$
$$+ (1 - q_{crit})[U(1, 1) - U(0, 1)] = 0$$

$$\iff P(A = 0 \mid do(B = 0))[q_{crit}\beta(0) - q_{crit}\beta(1)] + q_{crit}\beta(1) +$$
$$+ U(1, 1) - U(0, 1) - q_{crit}U(1, 1) + q_{crit}U(0, 1) = 0$$

$$\iff P(A = 0 \mid do(B = 0)) = \frac{(1 - \frac{1}{q_{crit}})(U(1, 1) - U(0, 1)) - \beta(1)}{\beta(0) - \beta(1)}$$

Which corresponds to Equation 6. We can compute $\beta(0) = -1$, $\beta(1) = 1$, $U(1,1) = 1$, and $U(0,1) = 0$, then:

$$P(A = 0 \mid do(B = 0)) = \frac{(1 - \frac{7}{5}) - 1}{-2} = 0.7 \tag{53}$$

Which is consistent with the CID and CPT shown in Figure 7.

**Step 5**: We can repeat the same procedure for $\sigma_1 = do(B = 1)$. In this case, since $A$ is binary we can compute $P(A = 1; \sigma_0) = 1 - P(A = 0; \sigma_0)$, in general we can repeat the procedure for all possible instantiations $a$ of $A$ to obtain $P(A = a; \sigma)$ for all $\sigma \in \sigma_B$.

**Step 6**: Since $P(A = 0|do(B = 0)) \neq P(A = 0|do(B = 1))$ (line 13), we can conclude that $B$ is a parent of $A$. In the general case, this approach ensures that $B$ is not just an ancestor but indeed a parent of $A$, because the intervention blocks all other paths from $B$ to $A$. Thus, if $B$ were not a parent, $P(A = 0|pa_A, do(B = 0))$ would be equal to $P(A = 0|pa_A)$. Once we have identified the set of parents, we can reconstruct the CPT for $A$ using the interventional distributions, e.g.: $P(A = 1|do(B = 0)) = P(A = 1|B = 0) = 0.3$ (lines 14 and 15).

As expected, this process allows us to learn both the correct graph structure and the CPT for $A$ (and, more generally, for all chance nodes that are not children of $D$).

## E.2  Example 2 - Multi-agent environment

Examine the multi-decision CID in Figure 8. It represents a cooperative game between two agents, agents A and B, respectively controlling decision variables $D_A$ and $D_B$. Both agents aim to maximize a shared utility function $U$, and operate in different contexts defined by the parent sets of their decision nodes ($Pa_{D_A} = \emptyset \neq \{Z\} = Pa_{D_B}$). As usual, we assume knowledge of the children for the decision node $D_A$, their CPTs, their parents, and the utility function associated with the node $U$. In this section, we demonstrate how to apply LearnCID to this example by providing a high-level description omitting the details we provided in the single-agent case.

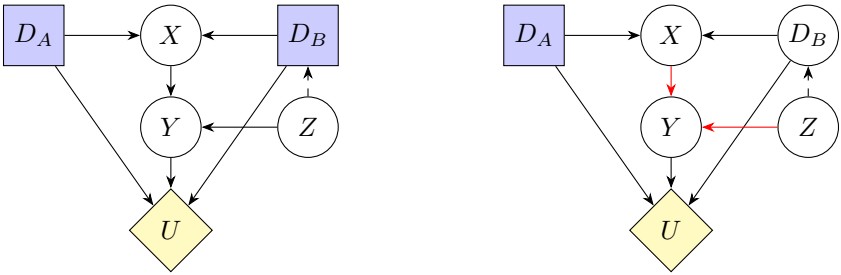

Figure 8: A multi-decision CID that represents an environment where agents A and B cooperate to maximize the utility $U$. The edges marked in red are unknown. Example 2 demonstrates how to adapt this CID to apply Algorithm 1 and recover the missing edges and CPTs for chance nodes.

Let $\Pi_\Sigma^*$ be the optimal policy oracle for $D_A$ and $\pi(D_B \mid Z)$ be any given policy that governs $D_B$.

- The nodes for which we need to learn the parents are $Y$ and $Z$.
- Node $Y$'s potential parents are $X$ and $Z$, whereas node $Z$'s only potential parent is $Y$.

We know $X$ cannot be a parent of $Z$ because by Assumption 4 we know $D_B$ is a parent of $X$, and we assumed to know the parents of every decision node. Therefore we know $X$ is a descendant of $Z$ meaning it cannot be its parent. Using Algorithm 1, we determine parental relationships as follows:

1. We check if $Z$ is a parent of $Y$. We consider the instantiation $Y = 0$. With Algorithm 1 we can compute $P(Y = 0 \mid pa_Y; \sigma_0)$ and $P(Y = 0 \mid pa_Y; \sigma_0')$ using $\sigma_0 := do(X = 0, Z = 0)$, and $\sigma_0' := do(X = 0, Z = 1)$ respectively. We observe that these two probabilities are equal. We repeat this process with $\sigma_1 := do(X = 1, Z = 0)$ and $\sigma_1' := do(X = 1, Z = 1)$, and again, $P(Y = 0 \mid pa_Y; \sigma_1) = P(Y = 0 \mid pa_Y; \sigma_1')$. Performing the same procedure

for $Y = 1$, we find that all pairs of interventions yield the same probabilities. Therefore, $Z$ is not a parent of $Y$.

2. Next, we check whether $X$ is a parent of $Y$. Comparing $P(Y = y \mid pa_Y; \sigma_0)$ with $P(Y = y \mid pa_Y; \sigma_1)$, and $P(Y = y \mid pa_Y; \sigma_0')$ with $P(Y = y \mid pa_Y; \sigma_1')$ for all $y \in \{0, 1\}$, we find that at least one of these pairs of probabilities differs. This confirms that $X$ is a parent of $Y$.

3. Finally, we consider $Z$ and aim to determine whether $Y$ is a parent of $Z$. According to the algorithm, we would again verify that the conditional probabilities remain unchanged when setting different values for $Z$. However, we can arrive at the same conclusion also by noticing that since $Y$ was found to be a child of $X$, $Y$ can not be a child of $Z$ because this would introduce a cycle in the graph. $Z$ has no other potential parents and therefore we have learned the full CID.

The fact that agents $A$ and $B$ cooperate on the same utility function does not affect the outcome. We would have obtained the same result even if agent $B$ was competing with agent $A$, or if agent $B$ did not try to optimize $U$ but was still able to causally influence it.

## F    Broader Impacts

This paper presents foundational research on the relationship between an agent's ability to adapt to distribution shifts and its causal understanding of its environment. Our findings contribute to a deeper understanding of what decision-making systems, including AI systems, must learn to be robust to distribution shifts. By characterizing the internal representations that such systems develop, our work promotes greater transparency in how decisions are made. This understanding lays the groundwork for designing more explainable, interpretable, and robust decision-making systems. In the long term, these advancements stand to benefit not only the users of these systems but also the broader population affected by their outcomes. If used maliciously, this technology could potentially be employed to probe the causal representations of decision-making systems that are intended to remain private. However, given the theoretical nature of our results and, in particular, the absence of a scalable algorithm to obtain full causal models, such misuse remains very unlikely in practice.

