# OpenReview forum: "Agents Robust to Distribution Shifts Learn Causal World Models Even Under Mediation"
_NeurIPS.cc/2025/Conference — NeurIPS 2025 poster_

### Official Review · Reviewer_dEGV · 2025-06-20

**Clarity:** 2
**Significance:** 4
**Originality:** 3
**Rating:** 3
**Confidence:** 2

**Summary:**

The paper provides theoretical results to show that decision-making robust to distribution shifts in the environment implies the learning of a causal model even under mediation, that is, when the decisions taken influence the environment, subject to a number of assumptions. The manuscript shows this by deriving an algorithm that learns the causal influence diagram (CID) and the corresponding conditional probability tables (CPT) from a base CID with some sets of already known information. The paper then extends the algorithm to a sequential decision-making case, by considering an unrolled partially-observable Markov decision process (POMDP) and extending their algorithm with a method to learn a CID from the POMDP.

**Questions:**

I have given a range of questions with context above, but for ease of access, I copy them below:
- Do you think your work can really be characterised as being about agents?
- Does your extension to POMDPs really reflect sequential decision-making? In particular
  - How does the discount factor come into play in Algorithm 2?
  - Is there any way this algorithm can model temporal relationships besides encoding everything in the state and relying on the Markov assumption?
  - Is there a notion of a goal here, e.g. can $U_t$ correspond to a long-term sparse reward signal?
- Can you really not use causal discovery algorithms, as you are taking some sort of samples in Algorithm 1 from the oracle
- In Algorithm 1, how are the edges of the CID $V$ updated to give $E'$?
- How do we or can we know that the base CID assumed in A2 is correct to begin with?
  - If it is correct, then doesn't that already imply that agents have a causal model of the world to begin with?

**Ethical Concerns:**

["NO or VERY MINOR ethics concerns only"]

**Final Justification:**

The paper presents a novel result regarding the learnt causal representations of robust agents by relaxing the no-mediation assumption of Richens and Everitt (2024). While the result seems significant, there is a great need for improved clarity in the paper regarding definitions, claims, limitations, and scope. The paper is also trying to address the case of robust agents in POMDPs and multi-agent settings, but these sections are not rigorously treated and feel more like an afterthought rather than the main contribution.

I believe the authors should focus on the main claim of the paper and strengthen this result, e.g. by improving the clarity around their definitions, the connections between those definitions, their relationship to Richens and Everitt, and by showing, with simulated data, that their algorithm works. As such, I think the paper is currently requiring significant revisions, but I encourage the authors to address the reviews and strengthen their paper.

**Limitations:**

I think the paper could benefit a *lot* from a more thorough discussion of its limitations. Throughout reading the paper, I was under the impression that the claims made in the introduction are not fully justified by the results, and in some cases, could potentially be construed as misleading. I do think the results are really valuable, but their interpretation would be much easier if the implied claim that all agents must be learning a CID were put into context by, e.g. providing an accessible (intuitive or non-formal) summary of the assumptions made in the paper.

I also thought that the discussion of related work as it relates to the paper's motivations was not explored to a satisfying extent. The work cites many related papers but doesn't provide a strong case for why CID-style causality would be the best form of causality to explore in this case. To strengthen the paper, it would be nice to see some examples mentioned in the introduction on how causal modelling addressed the challenges of distribution shift, reliable generalisation, and transparent decision-making in practice.

**Paper Formatting Concerns:**

None.

**Quality:**

4

**Strengths And Weaknesses:**

*Note: I want to flag that I am not an expert in "Pearlian" causal modelling and did not check the equations for correctness. My comments are, therefore, more high-level, and I welcome clarifications from the authors if I have misunderstood something regarding the technical aspects of the work.*

Overall, I thought the work was really interesting and, if correct, the contributions seem potentially valuable to understanding the nature of agency. However, I felt there was a disconnect between the motivation of the paper, its contributions, and related work, and I am hoping the authors could clarify some points in the discussion phase regarding their work's significance.

One of the main points I kept revisiting while reading the manuscript is whether the paper really deals with agents. Putting aside the debate on what it means to be an agent and have agency, the main argument of the paper seems to rely on the fact that we are in a one-shot decision-making setting with optimal decisions derived through a relative ordering defined over fixed utilities, as represented by the differences in Eq. 5 and 6. The authors also give an example of a sprinkler as an AI system, which I am not too sure about. In both cases, I am not convinced we are talking about agents, let alone AI. There is no mention of actions, goals, adaptability, or time. In this regard, I do appreciate the authors' extension of their algorithm to POMPDs, which seems to relax some of these assumptions. My concern is whether this extension violates any existing assumptions or introduces new ones (e.g. How does the discount factor come into play in Algorithm 2? Is there any way this algorithm can model temporal relationships besides encoding everything in the state and relying on the Markov assumption?)

On a technical note, I am wondering whether causal discovery algorithms could really not be used in the context of the manuscript. You are taking some sort of samples in Algorithm 1 from the oracle (though it is not very well explained in the text), so why not just use an existing algorithm? I am also wondering where in Algorithm 1 are the edges of the CID $V$ update to give $E'$?  Algorithm 1 also seems to assume that we have prior knowledge about the causal structure, and Assumption 2 adds quite a bit of prior information on its own. My concern here is: how do we know that the base CID is correct to begin with? And if it is correct, then doesn't that already imply that agents have a causal model of the world to begin with? This seems like assuming the conclusion in some ways.

Another note is that in current foundational agent discussions (on this see e.g. [Abel et al.](https://arxiv.org/pdf/2502.04403), [Kasirzadeh and Gabriel](https://arxiv.org/pdf/2504.21848), [Bowling and Elelimy](https://arxiv.org/pdf/2504.08161)), generalisation emerges as an important property of continual learning. This work necessarily glosses over this aspect of agency; otherwise, I reckon it would need a few hundred pages of extra content, but I found it curious that there was no mention of it. If the main claim of the paper ("agents robust to distribution shifts learn causal models...") is read at face value, then one might be tempted to assume that agents learning continually (or just learning in general) also have a sort of adapting causal graph in their internals, which seems rather strong in light of the fact that the paper makes a range of assumptions and simplifications. For clarity and expectations, it would be useful to include some caveats of the manuscript's results, if not in the title, at least in the introduction. Otherwise, others might cite the work as proof of the titular claim in a real-world setting or in places where the work's claims do not hold.

I also fail to fully understand the implications of the paper's results for future and related work. The motivations seem strongly based on the case that AI safety requires understanding whether and how agents form world models, but the discussion provides no insights into how the work ties in with, e.g. control. Specifically, what if we try to steer agents by trying to learn their causal model and intervene on their decision nodes, only to find out that most assumptions of the manuscript do not hold in practice, and there is actually no one causal model to steer or decision node to steer? I think acknowledging the limitations of the paper and how its results are reflected in the (implied) claim that all agents must learn a causal model would have gone a long way in presenting this work in a more grounded light.

---

> ### Author Rebuttal · Authors · 2025-07-31
>
> Thank you for your review.
>
> **Q1: “Do you think your work can really be characterised as being about agents?”**
>
> We appreciate the reviewer’s comments on the notion of agency. In this work, we implicitly adopted the definition from “Artificial Intelligence: A Modern Approach” by Russell and Norvig (1995), where an agent is any entity that maps percepts to actions to maximize expected utility. Under this definition, even simple systems such as smart sprinkler controllers qualify as agents, provided they map percepts (month of the year) to actions (activation of the sprinkler) and they maximize expected utility (keeping the lawn humid while minimizing water consumption). This broad definition is a strength as it enables our results to apply across a wide range of systems. We will clarify this point in the revision to avoid confusion about the intended scope of the term agent and to set appropriate expectations about the claims.
>
> **“[...] the main argument of the paper seems to rely on the fact that we are in a one-shot decision-making setting [...]”**
>
> The LearnCID algorithm is defined for environments that can be described with finite-dimensional CIDs. However, this does not imply that it can only be applied to one-shot decision tasks. It can also be applied to finite-horizon sequential decision tasks using similar techniques to those described in Section 3.2 for multi-decision CIDs and, as we have shown in Section 3.3, it can be utilized to recover the causal structure and the state-transition function of infinite-horizon sequential-decision tasks modeled with POMDPs through Algorithm 2.
>
> \
> **Q2a: “Does your extension to POMDPs really reflect sequential decision-making? In particular: How does the discount factor come into play in Algorithm 2?”**
>
> Thank you for pointing this out, as it allows us to highlight an important detail. Note that in Section 3.3, we use two different causal representations. First, we describe a POMDP using an infinite CID, where each utility node $U_t$ corresponds to the reward at time $t$. The goal is to find sufficient conditions under which LearnCID can be applied to recover the full causal structure and the CPTs, which we can then use to compute the state-transition function. Importantly,  all of these are **independent of the discount factor**.
> With Algorithm 2, we show that as long as time-homogeneity holds and the number of variables in the POMDP for a single timestep is finite, we can map this infinite CID to a finite CID that includes two consecutive timesteps of the infinite CID.
> This is the crucial step: to run LearnCID on the CID generated by Algorithm 2 we use an optimal policy oracle that for any given distribution shift returns a policy that maximizes the expected immediate reward. Together with the other assumptions, this is sufficient to recover the full causal structure of the original infinite CID describing the full POMDP and its state-transition function, **regardless of the original discount factor**. We will clarify this in the “Learning intra- and inter-temporal causal relationships” paragraph in Section 3.3, to make this distinction and the role of the discount factor clear.
>
> **Q2b: “Is there any way this algorithm can model temporal relationships besides encoding everything in the state and relying on the Markov assumption?”**
>
> Yes, CIDs can model sequential decision tasks for which the Markov assumption does not hold. Suppose we have a discrete time $t$, $V_t$ the set of variables at time $t$ (including the decision). Let $k>1$ be an integer, then if there exists an edge between a node corresponding to a variable in $V_t$ and a chance node corresponding to a variable in $V_{t+k}$, the sequential decision task modelled by this CID does not satisfy the Markov assumption. If the total number of nodes is finite, then, if the assumptions in the paper hold, we can run LearnCID and it would terminate returning all the causal relationships and the CPTs of the ancestors of the utility node. If the size is infinite but there is a structure that is periodically repeated, we can still apply LearnCID in a similar way to how we apply it to POMDPs even if the Markov assumption does not hold.
>
> **Q2c: “Is there a notion of a goal here, e.g. can $U_t$ correspond to a long-term sparse reward signal?”**
>
> In our framework, goals are represented by utility functions. As long as the CID produced by Algorithm 2 satisfies the LearnCID assumptions, the utility is allowed to correspond to a sparse reward signal, taking the value $0$ for most states and nonzero values only for a limited subset of states. In particular, the utility function must satisfy A9, which states that there does not exist a decision $d$ that is optimal regardless of the instantiation of the other variables that are needed to compute the utility (other parents of $U_t$). This assumption can be satisfied by sparse rewards and can be verified by evaluating the reward function for different combinations of its arguments.
>
> As an example, consider an environment with one variable “agent’s position” that takes values in the set {$1,2,3,4,5$}, the agent can move right (+1) or left (-1) and it is rewarded if it moves into 3. Therefore the sparse reward $\mathcal R(d,pos)$ is defined such that $\mathcal R(right,2)=\mathcal R(left,4)=1$ and $0$ otherwise. In this case $d=right$ is not optimal for $pos=4$ and  $d=left$ is not optimal for $pos=2$, therefore A9 is satisfied.
>
> \
> **Q3: “Can you really not use causal discovery algorithms, as you are taking some sort of samples in Algorithm 1 from the oracle”**
>
> Thank you for mentioning this, it is important to remark that standard causal discovery methods cannot be used in this setting. The procedure you refer to is part of $ALG_{q_{crit}}$ (Algorithm 1 in Richens and Everitt, 2024). We do not describe it in detail because we adopt it directly from their work. In that algorithm, the optimal policy oracle is repeatedly queried under different interventions (distribution shifts) produced by sampling the convex combination coefficient of a mixture of local interventions from a uniform distribution. Then the optimal policies returned by the oracle are used to compute the CPTs. Note that the oracle only provides policies for the decision node, and we do not obtain samples of any of the variables in the process. Standard causal discovery algorithms such as PC (Spirtes & Glymour, 1991), FCI (Spirtes, 2001), LiNGAM (Shimizu et al., 2006), and RESIT (Peters et al., 2014) require access to samples from the joint distribution, which is not the case in our settings.
>
> \
> **Q4: “In Algorithm 1, how are the edges of the CID $V$ updated to give $E’$?”**
>
> The set of edges is updated on line 13 when $Z_i$ is marked as a parent of $X$, meaning that the edge $Z_i \rightarrow X$ is in $E’$. We agree that not mentioning $E’$ explicitly may be confusing, we will make this clearer in the revision.
>
> \
> **Q5: “How do we or can we know that the base CID assumed in A2 is correct to begin with?”**
>
> A2 states that the partition of the nodes $V$ into $D$ (decision nodes), $U$ (utility nodes), and $C$ (chance nodes) is known. If you have doubts about assuming the knowledge of $V$, please refer to our response to W1 of reviewer **NWB8**. Once $V$ is specified, the partition follows naturally: each utility function, which we assume to be fully specified (A7), corresponds to a utility node forming the set $U$. The decision nodes $D$ represent agent outputs (e.g. the predicted label in an image classification task). All remaining variables constitute the chance nodes $C$.
>
> In general, there are several ways to test causal models. Causal models like CIDs encode conditional independence relationships between variables, which can be recovered from the causal graph using the concept of d-separation (Pearl, 2009). These relationships are testable and can be used to reject a model. Other techniques include randomized controlled trials, A/B tests, and comparing the model with domain knowledge.
>
> **Q5a: “If it is correct, then doesn't that already imply that agents have a causal model of the world to begin with?”**
>
> No. In this context, CIDs are simply a tool for jointly representing causal knowledge and decision-making tasks. The paper does not claim that robust agents have an internal implicit CID that they use to perform causal reasoning. Rather, robust agents encode the information needed to recover the CID, which serves as a tool for us to capture and visualize the causal knowledge we elicit from them. All five assumptions about the CID structure, including A2, make no claims about the agents’ internal representations of the world. Their role is solely to specify the conditions and clarify what must be known in advance about the CID representation of the decision-making task and the environment to apply LearnCID to extract the agent’s causal knowledge.
>
> \
> **Limitations:** \
> **“I do think the results are really valuable, but their interpretation would be much easier if the implied claim that all agents must be learning a CID were put into context [...]”**
>
> While we believe the paper is already transparent about its assumptions, we agree that readers focusing only on the introduction may misinterpret the results. To address this, we will add more context when presenting our results by also providing an intuitive description of the assumptions, helping ensure the contribution is properly framed and minimizing the risk of overgeneralization or misrepresentation in subsequent citations.
>
> Why opting for **CID-style causality?**
>
> CIDs build on results from causal inference and decision theory, particularly in complex scenarios like multi-agent environments and sequential decision-making. In such contexts, it is harder to represent the interplay between actions and outcomes with standard causal graphs. We will expand the discussion to better articulate these advantages and clarify why CIDs are a natural choice for our setting.

---

> > ### Comment · Reviewer_dEGV · 2025-08-01
> > **Some futher comments**
> >
> > Thanks for your clarifications, which were very helpful for a better understanding of the significance and limitations of the paper. Having read your responses to the other reviews as well, I am more confident that the paper can be a good contribution to the agent foundations literature and NeurIPS.
> >
> > I really appreciate your response to Q5a, and I think that this point should be much more salient, so that the reader knows exactly what is being represented and what is not. It doesn't help here that the paper repeatedly writes sentences such as:
> > 1. L7: 'agents capable of adapting to distribution shifts must learn the underlying causal relationships' - seems to imply that agents have an internal causal relationships;
> > 2. L37: 'must learn the causal structure of their environment' - seems to imply that there is a clear structure to this learned representation;
> >
> > I also had one follow-up question. How do you envision this line of work (agent foundations) addressing or contributing to the actual challenges of real-world systems, especially those that rely on current advanced AI tools, and not e.g. a sprinkler? There has been ample work in this space that is trying to formalise various aspects of agents (cf. almost anything from LessWrong), but it often seems to me that these works are more just displays of mathematical prowess rather than things people will then build on.

---

> ### Author Response · Authors · 2025-08-06
>
> Thank you for the feedback. We are glad our clarifications strengthened your confidence in the paper’s contributions.
>
> Regarding your follow-up question, first of all, we believe it is important to clarify that the sprinkler is only one of the examples we discussed, we also mentioned highly relevant real-world systems such as industrial robots, self-driving cars, and healthcare recommendation systems, which fall under the definition of agents we consider. We agree that sometimes one encounters exercises in mathematical formalism that end up having little or no impact on making progress on problems or addressing real challenges in AI. However, in this case, we believe that there are valid reasons justifying both the decision to work on this line of research and the use of the mathematical formalism found in the paper. For example, as we mentioned in the submission, this line of work opens the path to learning causal models from agents. In order to develop scalable algorithms for causal discovery from agents, the causal discovery community will need a solid mathematical framework that can provide strong theoretical guarantees, comparable to those offered by traditional causal discovery algorithms. Also, as AI systems are increasingly deployed and become more robust, having a way to learn causal models from them will expand the applicability of a wide range of techniques that rely on knowledge of a causal model, including all causal inference methods, counterfactual reasoning [1], and emerging fields like causal reinforcement learning [2]. There is also growing evidence of the use of causal AI In industry. Novartis applies causal engines to optimize clinical trials [3], IBM Instana uses causal models for root-cause analysis [4], and manufacturers like Lindt use causal AI to improve yield and reduce defects [5].
>
> Thank you for pointing out specific sentences that highlight the issue you previously mentioned. While these sentences are not incorrect per se, since the fact that we can construct the causal model from the robust agent implies that the robust agent must encode that causal knowledge in the first place, we confirm that without an appropriate context in the introduction they can be misinterpreted (e.g. for L37 “causal structure of their environment” is meant w.r.t. the fact that the environment is represented with a CID, not meaning that the causal knowledge the robust agent possesses is structured in any specific way). We confirm we will revise the introduction to provide a clearer framing from the very beginning of the paper to put these and similar sentences into context.
>
> [1] Causality, Pearl, 2009 \
> [2] Towards causal reinforcement learning, Bareinboim et al., 2024 \
> [3] Interview with Bülent Kızıltan, Head of Casual & Predictive Analytics, AI Innovation Center at Novartis. Using AI and Data  Science to Lead Advancements in Medicine - Redis \
> [4] Causal AI-based Root Cause Identification: Research to Practice at Scale, Jha et al, 2025 \
> [5] Index Ventures backs AI manufacturing startup EthonAI in $16 mln funding round, Coulter, 2024

---

> > ### Comment · Reviewer_dEGV · 2025-08-06
> >
> > Thanks again for the clarifications. The additional context for why this line of work is relevant has been very helpful.
> >
> > I think this work has merit and does constitute a novel contribution to the field. The authors have been helpful in clarifying some of my questions, but having read through the other reviews, there seems to be a significant need for clarity, better definitions, better discussion of scope, and clarity over limitations.
> >
> > I agree that NeurIPS is definitely suitable for theoretical work like this, but I would also note that the Richens and Everitt paper has also performed some empirical demonstration of their work. Perhaps instead of trying to cram both POMDPs and the multi-agent setting into the paper, it might be more worthwhile to explore the ramifications of the main results of the paper in more depth (perhaps by discussing parts of the appendix in the main paper), present some simulated experiment, and show why the mediation setting is so important to address, perhaps even including a simple, illustrative example to demonstrate how relaxing the no-mediation assumption makes a practical difference. As such, I am inclined to keep my score as is.

---

> > > ### Author Response · Authors · 2025-08-08
> > >
> > > Thank you again for your continued engagement. Regarding the point of empirical validation, we’d like to note that the Richens and Everitt paper includes experiments only for the approximate case, not for the exact setting we address in our work.
> > >
> > > Regarding scope, we included both the multi-agent and sequential extensions to illustrate the broad applicability of our core theoretical insights. The results presented in these settings are applications of the LearnCID algorithm, which is the central contribution of the paper. We recognize that balancing depth and breadth is always a challenge. To manage this, we followed the common practice of placing full proofs in the appendix to keep the main text focused and accessible. We appreciate your suggestions and thank you for them.

---

### Official Review · Reviewer_NWB8 · 2025-07-04

**Clarity:** 4
**Significance:** 3
**Originality:** 4
**Rating:** 5
**Confidence:** 3

**Summary:**

The paper proposes an algorithm and proves it's consistency for discovering causal world models from robust agents when they affect the environment (mediated setting). The algorithm is further applied in the multi-agent setting and POMDP setting. It expands the work from Richens and Everitt [1].

The implications of this work are that robust optimal policies need to learn the CID in order to be robust to distribution shifts in the mediated setting.

[1]  Robust agents learn causal world models, ICLR 2024

**Questions:**

## Questions

l.99 - this sounds more like a theorem or proposition than a definition since it makes a claim about existance of optimal policies in that a re different for CIDs statisfying the conditio?  This cannot be a definition. A follow-up to this question is how does this relate to the notion of identifiability?

l.112 - the way the intervention is written is really easy to confuse with the hard intervention case, but in effect the "local" intervention is introducing a transformation of the random variable X? So f is a push-forward map, correct? Shouldn't this just be seen as introducing a "mechanism" that introduces another ancestor Y to children of X, i.e. we have a a dependance chain introduced,  X ->  Y -> children(X). Might be simpler to explain it this way.

**Ethical Concerns:**

["NO or VERY MINOR ethics concerns only"]

**Final Justification:**

The authors have answered all of my questions and I think that the paper makes a solid technical contribution in the realm of causal world models and decision making under distribution shifts. I am not increasing my score because of the lack of experimental evaluation and significance, I still think it's good work.

**Limitations:**

yes

**Paper Formatting Concerns:**

no concerns

**Quality:**

3

**Strengths And Weaknesses:**

## Strengths

Several theoretical insights considering causal discovery of the underlying graphical model from agents robust to distribution shifts in the mediated setting, multi-agent setting and POMDP setting.

The setting of inferring the causal model without the assumption that the agent is not influencing the environment is novel as far as I know.


## Weaknesses


The algorithm has similar limitations as classical causal discovery algorithms, it works only in the structured setting where we have nice separation of the variables. The more interesting setting that requires discovery of the latent variables themselves is not tackled.

There is no experimental validation of the algorithm in any interesting setting for the ML community.

There is no complexity analysis of the algorithm regarding runtime.

---

> ### Author Rebuttal · Authors · 2025-07-31
>
> Thank you for your review and your observations. Regarding your questions:
>
> **Q1: “l.99 - this sounds more like a theorem or proposition than a definition since it makes a claim about existance of optimal policies in that a re different for CIDs statisfying the conditio? This cannot be a definition. A follow-up to this question is how does this relate to the notion of identifiability?”**
>
> Thank you for bringing this to our attention. We would like to update it as follows: \
> "$\underline{\text{A CID }M \text{ is said to satisfy domain dependence if}}$ there exist $P(C = c)$, $P^\prime(C = c)$, both compatible with the CID $M$ such that $\pi^* = argmax_\pi\mathbb E_{P}^\pi[U]\implies \pi^*\not = argmax_\pi\mathbb E_{P^\prime}^\pi[U]$.” \
> We look forward to the reviewer's comments on this.\
> Regarding identifiability, under the assumptions made in the paper, if domain dependence does not hold, then it is not possible to learn the causal model from the policy oracle. Intuitively, the reason is that if domain dependence does not hold, there exists a policy whose optimality is unaffected by any distribution shift. In this case, the optimality of a decision system is not affected by changes in the environment, and causal knowledge becomes unnecessary.
>
> \
> **Q2: “l.112 - the way the intervention is written is really easy to confuse with the hard intervention case, but in effect the "local" intervention is introducing a transformation of the random variable X? So f is a push-forward map, correct? Shouldn't this just be seen as introducing a "mechanism" that introduces another ancestor Y to children of X, i.e. we have a a dependance chain introduced, X -> Y -> children(X). Might be simpler to explain it this way.”**
>
> We agree that the notation doesn’t make it immediately clear whether the intervention we are applying is hard or not. For these results, we decided to stick with Richens and Everitt’s notation. Yes, f is indeed a push-forward map, and the idea you proposed to add an intermediate node between the intervened variable X and its children is correct and can be an effective way to make the concept more intuitive and easier to understand. Thank you for the suggestion.
>
> About the weaknesses that were pointed out:
>
> **W1:** \
> **“The algorithm has similar limitations as classical causal discovery algorithms, it works only in the structured setting where we have nice separation of the variables. The more interesting setting that requires discovery of the latent variables themselves is not tackled.”**
>
> Thank you for this insightful comment. We agree that the setting where latent variables must be discovered directly is interesting. As we understand it, the issue being raised is that the paper assumes knowledge of the latent variables and also assumes causal sufficiency, meaning there is no latent confounding. As you also noted, assuming causal sufficiency is a common approach in the causal discovery literature, especially when proposing new methods to address the problem. Examples include LiM (Zeng et al. 2022), RESIT (Peters et al., 2014), LiNGAM (Shimizu et al., 2006), and PC (Spirtes and Glymour, 1991).
> Regarding the assumption of knowing the set of latent variables, the problem of identifying latent variables and the problem of discovering causal relationships between them are, in principle, separate. There is active research focused on learning latent representations, including latent variables, for causal models. The review suggests that you are probably familiar with techniques like those of causal representation learning that among other things aim at learning the latent variables for causal models from high-dimensional data. All of this is important and interesting work, which we regard as complementary rather than overlapping. A discussion of how these techniques could be integrated with the paper’s framework would be a valuable addition, and we will highlight these possibilities in the revision.
> It is also important to note that both of these assumptions, along with the others in this paper, were either explicitly stated or implied in Richens and Everitt (2024). In fact, this work relaxes their setting.
>
> **W2:** \
> **“There is no experimental validation of the algorithm in any interesting setting for the ML community.”**
>
> Thank you for emphasizing the importance of empirical validation and we fully appreciate it.
> As we state in the submission, our work is purely theoretical in nature, aligning with the types of contributions NeurIPS explicitly supports. There is a strong tradition at NeurIPS of high-impact theoretical work without empirical experiments, such as: “A Universal Law of Robustness via Isoperimetry” by Bubeck and Sellke (NeurIPS 2022), “Causal Discovery from Soft Interventions with Unknown Targets: Characterization and Learning” by Jaber et al. (NeurIPS 2020), and “Transportability from Multiple Environments with Limited Experiments: Completeness Results” by Bareinboim and Pearl (NeurIPS 2014).  That said, we agree that empirical validation is a natural next step, but only after the foundational theory is well established. Our work aims to lay that foundation, which future empirical efforts can build upon.
>
> **W3:** \
> **“There is no complexity analysis of the algorithm regarding runtime.”**
>
> As mentioned in Section 3.1 line 237, we discuss the runtime complexity of LearnCID in Appendix A.3, starting on line 667.

---

> > ### Comment · Reviewer_NWB8 · 2025-08-09
> > **Thank you for your response**
> >
> > I thank the authors for their response and addressing my concerns. I will argue for the acceptance of the paper. While I still think that more meaninfgul experiments would make the work stronger, I appreciate the theoretical contributions.

---

### Official Review · Reviewer_iDtz · 2025-07-05

**Clarity:** 2
**Significance:** 2
**Originality:** 2
**Rating:** 3
**Confidence:** 5

**Summary:**

Past work shows that agents robust to distributional shifts learn a causal model of their environment. However this result assumes that the agent does not influence its environment with its actions (“no mediation”). This paper extends the previous result to the setting with mediation. The paper presents an algorithm for inferring a causal influence diagram from an optimal policy oracle, and discusses its application in multi-agent and sequential environments.

**Questions:**

Richens also proves approximate theorems based on regret bounded policy oracles. Do these results carry over to the mediation setting?

**Ethical Concerns:**

["NO or VERY MINOR ethics concerns only"]

**Final Justification:**

Overall I feel that the paper's formal work is not really tight enough, the novel contribution is modest, more could be done to make the paper intuitive and accessible, and the later parts are weak (e.g., multi agent extension). I think a purely theoretical paper should be more solid on all of these lines to be at the acceptance level for neurips.

I think a future version of the paper could make a solid contribution if it becomes more focused, tightens up the formal work, and perhaps adds experimental validation.

**Limitations:**

No substantial discussion of limitations. The authors should be more upfront about some assumptions (e.g., in the multi agent discussion, the assumption of shared utility)

**Quality:**

2

**Strengths And Weaknesses:**

I appreciate what the paper is trying to do and think that extending the Richen's results is an important direction. However, the formal work in this paper is not quite tight enough for me, and the contributions are very modest.

Here are my comments:

Major.

- There is no related work section and several references are missing --- in particular I think it would be good to discuss the literature on CIDs ingrate detail, including their importance to problems in safe AI (e.g., discussing some of the papers from the causal incentives group https://causalincentives.com)
- There is also this new paper, from Richens, which extends the same Robust Agents results https://arxiv.org/abs/2506.01622 (I haven't read this paper and of course can't blame the authors for not having seen such a new paper, but it's still highly relevant)
- Aspects of the theory are not properly introduced, for example, policies and CPTs are not defined.
- The assumptions and main results are not explained clearly enough --- there should be more intuitive discussion of each to help the reader and make the paper more accessible.
- In a theoretical paper, you should include sketch proofs of theorems in the main paper rather than in the appendix.
- Some of the work is basically informal (e.g., the multi-agent section)

Minor.

The first paragraph of the introduction contains too much information. It should try to more concisely state the problem — lots of the detail here should be in related work imo.

The intro to CIDs isn’t that clear. The two weather examples dont immediately connect to the formal work. You should also start by saying, e.g., “CIDs are a type of probabilistic graphical model…” as of now you start talking about “the graph” without explanation. Further, definition 1 introduces a CID as a CBN but CBNs have not been defined or mentioned.

“showing that it is possible to recover the causal structure and the Conditional
Probability Tables (CPTs) of the variables describing the environment by observing the agent’s
optimal policies under distribution shifts.”

CPTs and policies and optimality not defined.

Terminology is often non-standard. Why depart from the literature? “CPTs” —> CPDs; “informational arcs” —> information links

“In this paper, the
terms adaptable agent and robust agent refer to an agent that maintains optimal performance across
all the possible distribution shifts in its environment.”  so these mean the same thing? Just stick to robust agent?

In CID graphs, edges into decisions (i.e., information links) should be dashed.


“Each agent may correspond to a different set of decision nodes and have access to a distinct subset of
observable variables.” A proper formal analysis of multi-agent settings requires e.g. Causal Games cf Hammond https://arxiv.org/abs/2301.02324

No need for policy oracles to be optimal by definition. (You already say in many places “an optimal policy oracle”.

Main results section should be split up into different sections.

“multi-agent systems represented with multi-
decision CIDs” I don’t think this is a valid model, at least it misses all the game theoretic dynamics. Even if the agents optimise the same utility (a serious limitation) you dont deal with e.g., different observations and memory.

---

> ### Author Rebuttal · Authors · 2025-07-31
>
> Thank you for your review. Starting from the question:
>
> **Q1: “Richens also proves approximate theorems based on regret bounded policy oracles. Do these results carry over to the mediation setting?”**
>
> We believe these results carry over to the mediation setting, though the formal work to confirm this has not yet been undertaken. The three contributions outlined in the first part of the paper, namely showing that domain dependence is no longer sufficient, establishing unidentifiability results for the children of the decision node, and developing a new procedure to recover the causal model in the optimal case, are prerequisites for extending the approximate results to the mediated case. Exploring the approximate case in the mediation setting is an important direction for future work.
>
> \
> **Limitations: “The authors should be more upfront about some assumptions (e.g., in the multi agent discussion, the assumption of shared utility)”**
>
> Thank you very much for pointing this out. In fact, it is not necessary for different agents to optimize the same utility function. Only one of the agents needs to optimize the utility function, while the others only need to be able to influence it. Specifically, it suffices that all decision nodes under consideration are ancestors of the utility node, and that we have access to a policy oracle for at least one of these decision nodes which is optimal with respect to the utility function associated with the utility node. The proposed methods for applying LearnCID to multi-decision CIDs can be applied to this situation without modification. We will update Section 3.2 to mention this.
>
> Regarding the observations.
>
> **Major 1:** \
> **“There is no related work section and several references are missing --- in particular I think it would be good to discuss the literature on CIDs ingrate detail, including their importance to problems in safe AI (e.g., discussing some of the papers from the causal incentives group)”**
>
> While our current focus has been on citing directly relevant peer-reviewed work, we will expand the related work section to include a more detailed discussion of the broader CID literature, including its relevance to safe AI. We believe we have cited the most relevant existing works, but we will carefully review the literature again (including those from the causal incentives group) and ensure that everything relevant is cited. Does the reviewer have any suggestions for papers we should include that we may have overlooked?
>
> **Major 2:** \
> **“There is also this new paper, from Richens, which extends the same Robust Agents results (I haven't read this paper and of course can't blame the authors for not having seen such a new paper, but it's still highly relevant)”**
>
> The paper was not cited because it was first published on June 2nd, and the full paper submission deadline for NeurIPS 2025 was on May 22nd. We have read this paper. It shows that an agent able to adapt to different goals must have learned a predictive model of the environment. The results of the paper are indeed interesting and the approach is similar. We will cite it in our revision. However,  the paper is not about causal models, causal knowledge, or distribution shifts in the environment, so it does not overlap with our work in any way.
>
> **Major 3 and 4:** \
> **3. “Aspects of the theory are not properly introduced, for example, policies and CPTs are not defined.”** \
> **4. “The assumptions and main results are not explained clearly enough --- there should be more intuitive discussion of each to help the reader and make the paper more accessible.”**
>
> Regarding policies and CPTs, we understand that accessibility is valuable, and we will take the necessary steps to address this.
> We also agree that understanding the assumption is very important. We will include a more intuitive description of the assumptions.
>
> **Major 5:** \
> “In a theoretical paper, you should include sketch proofs of theorems in the main paper rather than in the appendix.”
>
> Thank you for the suggestion. To clarify, the appendix contains only complete proofs. Similar to what they do in Richens and Everitt (2024), we chose to keep the main text focused on the high-level ideas, while providing full technical details in the appendix to maintain readability.
>
> **Major 6:** \
> **“Some of the work is basically informal (e.g., the multi-agent section)”**
>
> Could you please pinpoint which parts you found to be informal? We would greatly appreciate your feedback so we can address them thoroughly.
>
> \
> **Minor issues**
>
> About the minor issues, thank you very much for your corrections and suggestions in the introduction, we will address all of these and also add the CBN definition and turn the edges going into decision nodes from full to dashed. We also agree that it would be more appropriate and equally convenient to separate the notion of policy oracle and that of its optimality, we will update the definition.
>
> About CPTs vs CPDs. CPTs are standard terminology in discrete bayesian networks. The two terms are not equivalent. The reason why we used CPTs in this context is that while CPDs can be associated with both discrete and continuous variables, CPTs are only defined for discrete variables, which are those we consider throughout the paper.
>
> \
> **“ ‘multi-agent systems represented with multi- decision CIDs’ I don’t think this is a valid model, at least it misses all the game theoretic dynamics. Even if the agents optimise the same utility (a serious limitation) you dont deal with e.g., different observations and memory.”**
>
> You are correct that the mechanised MAID is a richer model when it comes to combining causal and game-theoretical reasoning by adding the decision rules and the parameters of the non-decision variables to the graph. Multi-decision CIDs are a special case of MAID in “Multi-agent influence diagrams for representing and solving games” Koller and Milch (2001), with edges that do not enter decision nodes representing causal relationships, and without explicitly specifying the set of agents and the correspondence between agents and decision nodes. We chose this minimal model because we only need a representation of multi-agent systems that captures the causal dependencies between the environment and decision variables both concisely and flexibly. We believe the results can be specialized for mechanized MAIDs with modest additional work.
>
>
> _Regarding memory and different observations_ \
> Still, multi-decision CIDs can represent agents with different observations by allowing each decision node to have distinct parent sets and by masking observations through interventions, which is the approach we adopt in this paper.  In a similar way, memory of past observations and decisions can be implemented by including the corresponding nodes as parents of the subsequent decision that relies on this memory.
>
> _On having multiple utility nodes_ \
> We agree that it would be interesting to develop a version of LearnCID for multiple utility functions, perhaps by using an oracle that returns a policy optimizing a weighted combination of utilities. However, observe that even in our settings, there can be more than one utility function, and different agents are allowed to optimize different utility functions. When we want to apply LearnCID, we just need to prune all the utility nodes except one and only keep the chance nodes that are its ancestors. We can repeat the same procedure for different utility functions in the same environment if we have at least one corresponding optimal policy oracle available. We will make this clearer by updating Section 3.2.

---

> > ### Comment · Reviewer_iDtz · 2025-08-03
> >
> > Thanks for your detailed response. I have skimmed the other reviews and tried to take them into consideration. Overall I am not really inclined to increase my score as I think the paper needs substantial improvement.
> >
> > > We believe these [approximation] results carry over to the mediation setting, though the formal work to confirm this has not yet been undertaken.
> >
> > Maybe it would be better to focus on these results, to completely capture the no mediated single agent setting, before moving on to more complicated multi agent extensions
> >
> > > Specifically, it suffices that all decision nodes under consideration are ancestors of the utility node, and that we have access to a policy oracle for at least one of these decision nodes which is optimal with respect to the utility function associated with the utility node.
> >
> > What's the notion of optimality here? In the multi-agent setting optimality isn't really defined without fixing the policies of the other agents (reducing to single-agent setting, which is how your proposals for using the LearnCID algorithm. work, right)  --- it's more fruitful to think about equilibria concepts in a full multi-agent setting imo. Without addressing the game-theory I think the multi-agent analysis will always be extremely limited.
> >
> > > related work
> >
> > It's also good to provide greater context for your work in relation to the literature. On specific references, see
> > https://arxiv.org/abs/2208.08345
> > https://arxiv.org/abs/2301.02324
> > https://arxiv.org/abs/2402.07221
> >
> >
> > > Major 2:
> >
> > Thanks, TBC I did not mean to highlight this as a limitation of your work, I just wanted to flag it as an important consideration
> >
> > > major 3 and 4
> >
> > I do think it's important to tighten things up here and improve readability so your paper can have greater impact
> >
> > > Major 5:
> >
> > maybe that's a reasonable approach
> >
> > > Major 6:
> >
> > For instance, "Now suppose we introduce a new agent A′into the same environment." What does this mean? a new decision? a new policy oracle?
> >
> > I'm pointing out that example after the fact, but I think my general feeling was just that lots of things hadn't been properly introduced or defined (as noted above) and the multi-agent extension was basically hand wavy and not really multi agent inn the important ways.
> >
> > > About CPTs vs CPDs. CPTs are standard terminology in discrete bayesian networks. The two terms are not equivalent.
> >
> > Thanks for clarifying
> >
> > On the multi-agent extension: I think doing this properly is a lot of work --- a full paper in itself. I think your paper might be better leaving that extension for future work and focusing on fleshing out the single agent case (eg with approximation results). It would be cool to have experimental validation too.
> >
> > I also want to flag that I think the theoretical extension to no mediation setting is pretty modest in itself and this is a reason for my low score. (basically the novelty is minor)

---

> ### Author Response · Authors · 2025-08-06
>
> Thank you for your feedback on the rebuttal and your new comments.
>
> **"What's the notion of optimality here? [...] it's more fruitful to think about equilibria concepts in a full multi-agent setting imo. Without addressing the game-theory I think the multi-agent analysis will always be extremely limited."**
>
> In Section 3.2, we provide sufficient conditions to apply LearnCID and recover the multi-decision CID representing a multi-agent system. As you say, the two proposed techniques require the policies to be fixed. Note that, as a special case, if the system is in an equilibrium, then all the policies for the other decision nodes are indeed fixed, and if the LearnCID assumptions hold, it is possible to learn the causal structure of the environment.
>
> Optimality for one decision node $D$ is once again captured with the notion of optimal policy oracle. We can map any policy of the secondary decision nodes to any compatible policy with a mixture of local interventions. For each such intervention, we expect optimal behavior from the agent. Therefore, the availability of an optimal policy oracle in this setting implies the presence of an agent that can achieve optimal performance (maximize expected utility) through decision node $D$ for every possible set of policies for the other decision nodes. Consequently, whether or not the system is in equilibrium becomes irrelevant. As long as the policies are faithful and the LearnCID assumptions hold, it is possible to recover the causal structure and the CPTs of the chance nodes of the original multi-decision CID. While mentioning the above in the multi-agent section could be interesting, applying the techniques proposed in the paper does not require discussing game-theoretic dynamics.
>
> It is possible to think about this as having a multi-agent system which is in a certain state (could be an equilibrium, but not necessarily), and then query the robust agent about what its behavior would be under specific shifts in the system (w.r.t. the current state) that potentially involve both the environment and the behavior of other agents. Note that querying the agent doesn’t require any interaction with the system itself. By asking these kinds of questions, it is possible to reconstruct the causal model.
>
> **"Major 6: For instance, "Now suppose we introduce a new agent A′into the same environment." What does this mean? a new decision? a new policy oracle?"**
>
> Thank you for the example. In this case, introducing an agent in an environment represented with a CID means introducing a new set of decision nodes associated with a new agent A’ and the edges involving these nodes. We’ll make sure to make this clear in the revision.
>
> **“I also want to flag that I think the theoretical extension to no mediation setting is pretty modest in itself and this is a reason for my low score. (basically the novelty is minor)”**
>
> We suppose you meant the extension to the **mediation** setting. As you know, the state of this line of work prior to this paper only covers the unmediated and one-shot single-agent (and single-decision) decision tasks. As we argued in the paper, these restrictions exclude most of the settings and problems that the community studies and the practitioners work with. From the results in Richens and Everitt (2024), it is not trivial that the framework can be generalized to mediated tasks, and the applicability of their results to these more general settings is questionable. From their paper’s review:
>
> - Reviewer **gS4b**: “The scope is currently limited to unmediated decision tasks. Extending the results to broader RL settings would increase applicability (although I acknowledge that seems significantly more challenging task and out of scope of this work - it’s just a personal curiosity at this point and would be excited to see the next paper already).”
>
> To which the **authors** replied:
>
> - “We tried to extend the results to unmediated [typo: they meant mediated] decision tasks, but unfortunately this was too challenging in the time available. We agree this is an important direction for future work.”
>
> For instance:
> 1. It is not trivial that in the mediated case, the set of children of the decision node cannot be identified (Theorem 3), and how to handle this in the algorithm.
> 2. Richens and Everitt assume domain dependence and show that it implies A9 in the unmediated case. As we discuss in Appendix C, this implication does not hold in the mediated case. This requires a different proof strategy, which allows us to prove the correctness of LeanCID (Theorem 2) under a strictly weaker set of assumptions than those used by Richens and Everitt (2024).
> 3. Theorem 2 covers the case for nodes in $Anc_U \cap Desc_D$, which is absent in Richens and Everitt's original work.
>
> With this paper, we want to show that this framework extends to a broad family of decision tasks, including multi-agent and sequential tasks, thereby encouraging the community to engage with and build upon it.

---

### Official Review · Reviewer_pmnc · 2025-07-08

**Clarity:** 2
**Significance:** 3
**Originality:** 2
**Rating:** 4
**Confidence:** 1

**Summary:**

This paper considers the decision-making with Mediation where the actions have an effect on the environment and develops a foundational framework to show that agents robust to distribution shifts must have learned a causal model.  The authors propose the LearnCID algorithm which learns Causal Influence Diagrams (CID). Then, the authors present the application of LearnCID to multi-agent environments and POMDPs.

**Questions:**

No further questions.

**Ethical Concerns:**

["NO or VERY MINOR ethics concerns only"]

**Final Justification:**

I remain the rating as the author rebuttal has addressed my concerns.

**Limitations:**

The authors discussed the limitations in the last section.

**Quality:**

3

**Strengths And Weaknesses:**

The paper has the following strengths.
+ The paper extends the existing results by removing the assumption of no mediation, which builds a foundation for more general use cases.
+ The paper theoretically show that the causal model is learnable.
+ The paper considers the concrete applications of POMDP and multi-agent systems.

The paper has the following limitations.
- The framework only applies to discrete variables, which may limit its application for general decision-making problems.
- The framework only focuses on learning the causal model but how to combine it with traditional data-driven RL algorithms is not discussed.
- Would it be possible to quantify the distribution shift and show the effect of the distribution shift on the performance?

---

> ### Author Rebuttal · Authors · 2025-07-31
>
> Thank you for your review.
>
> **“Would it be possible to quantify the distribution shift and show the effect of the distribution shift on the performance?”**
>
> The answer is yes to both. As discussed in Section 2 lines 104-125 and Appendix D, it is possible to formalize a distribution shift as an intervention on the causal model. In this case, once you obtain a CID, you can use it to compute an updated probability distribution for any given distribution shift. At that point, it is possible to quantify the shift by, for example, comparing the original probability distributions with those obtained after the shift. This can be done by computing the Wasserstein distance, the KL-divergence, or using any other technique to quantify the difference between probability distributions.
>
> As for measuring the effect of the distribution shift on performance, given a policy and a utility function, you can compute the expected value of the utility for that policy under a given distribution shift using the CID. Again, if the shift is in the form of an intervention, you can first update the probability distributions, and possibly the graph structure, and then use the updated distribution and the policy to directly compute the expected utility. This can be compared with the expected utility under no distribution shift for the same policy.
>
>
> **“The framework only focuses on learning the causal model but how to combine it with traditional data-driven RL algorithms is not discussed.”**
>
> Thank you for raising this point. In this work, we consider an agent that is already robust to distribution shifts, without making any assumptions about how the agent should be trained. This line of work is very novel, as, to the best of our knowledge, there are no other results outside of Richens and Everitt (2024) and this paper. For this reason, we decided to give priority to the development of the core concepts. That said, there are interesting opportunities to explore connections between this framework and RL algorithms. For example, integrating an algorithm like LearnCID into a data-driven RL method could enable a decision system to both learn the task and reveal the causal model simultaneously, or understanding how the use of an RL algorithm might influence the process of learning causal knowledge would also be insightful. These are significant topics and will become important directions for future work once the theoretical foundations of the core concepts have been fully developed.

---

> > ### Comment · Reviewer_pmnc · 2025-08-05
> >
> > Thank you for the clarification which helps me understand the paper.

---

### Comment · Area_Chair_JGzs · 2025-08-03
**Reviewers please respond to the rebuttal!**

Dear reviewers,

if you have not yet responded to the rebuttal of the authors, please do so as soon as possible, since the rebuttal window closes soon.

Please check whether all your concerns have been addressed!  If yes, please consider raising your score.

Best wishes,
your AC

---

### Decision · Program_Chairs · 2025-09-17

**Decision:**

Accept (poster)

**Comment:**

The paper significantly extends the work of Richens and Everitt (2024) to the mediation settings and provide and work out the necessary theory.  Furthermore they provide an algorithm for mediated single-agent scenarios and multi-agent environments.  While not all reviewers are convinced I found many of the critiques of the negative ones somewhat fluffy and not really helpful.  On the contrary, the rebuttal of the authors are quite convincing, so I suggest to accept this paper!